# SCARB2 drives hepatocellular carcinoma tumor initiating cells via enhanced MYC transcriptional activity

Feng Wang[1,6], Yang Gao[1,2,6], Situ Xue[1,6], Luyao Zhao[1], Huimin Jiang[1], Tingting Zhang[1], Yunxuan Li[1], Chenxi Zhao[1], Fan Wu[3], Tana Siqin[3], Ying Liu[1], Jie Wu[1], Yechao Yan[1], Jian Yuan [4,5] ✉, Jian-dong Jiang [1] ✉ & Ke Li [1] ✉

CSCs (Cancer stem cells) with distinct metabolic features are considered to cause HCC (hepatocellular carcinoma) initiation, metastasis and therapeutic resistance. Here, we perform a metabolic gene CRISPR/Cas9 knockout library screen in tumorspheres derived from HCC cells and find that deletion of SCARB2 suppresses the cancer stem cell-like properties of HCC cells. Knockout of *Scarb2* in hepatocytes attenuates HCC initiation and progression in both MYC-driven and DEN (diethylnitrosamine)-induced HCC mouse models. Mechanistically, binding of SCARB2 with MYC promotes MYC acetylation by interfering with HDCA3-mediated MYC deacetylation on lysine 148 and subsequently enhances MYC transcriptional activity. Screening of a database of FDA (Food and Drug Administration)-approved drugs shows Polymyxin B displays high binding affinity for SCARB2 protein, disrupts the SCARB2-MYC interaction, decreases MYC activity, and reduces the tumor burden. Our study identifies SCARB2 as a functional driver of HCC and suggests Polymyxin B-based treatment as a targeted therapeutic option for HCC.

Hepatocellular carcinoma (HCC) is the most common type of primary liver cancer and is characterized by both phenotypic and molecular heterogeneity[1]. The high mortality of HCC is caused by the lack of suitable biomarkers for early detection, the inadequate understanding of HCC heterogeneity, and therapeutic resistance[2]. Cancer stem cells (CSCs) of HCC are considered to be responsible for initiating heterogeneous tumor lesions and contributing to tumor relapse, metastasis, and therapeutic resistance[3,4]. Furthermore, chemotherapy and targeted therapy for HCC have been demonstrated to select for the outgrowth of therapy-resistant stem-like cancer cells, which facilitate tumor recurrence[5,6]. Currently, extraordinary progress has been made in the identification of liver CSCs on the basis of surface and cellular prognostic/diagnostic markers, such as cluster of CD44, CD90, CD133, CD24, and EpCAM in liver CSCs[7,8]. However, CSCs can also be identified in populations of cells not expressing these markers[9]. Thus, understanding the detailed regulatory mechanism of CSC emergence and expansion provides opportunities to improve the outcomes of patients with HCC.

The proto-oncogenic transcription factor MYC is deregulated in almost all human cancers, especially HCC, and high levels of MYC are associated with poor prognosis[10]. Previous studies have established the contribution of MYC to the pathogenesis of liver cancer[11,12] and demonstrated that activation of MYC is required for oncogenic reprogramming of adult hepatocytes into CSCs[13]. Thus, targeting MYC, especially in combination with traditional therapies, is considered an attractive therapeutic strategy for HCC. MYC is subject to a series of

[1]Institute of Medicinal Biotechnology, Chinese Academy of Medical Sciences & Peking Union Medical College, 100050 Beijing, China. [2]The Affiliated Cancer Hospital of Zhengzhou University & Henan Cancer Hospital, Zhengzhou 450008, China. [3]Cancer Hospital, Chinese Academy of Medical Sciences & Peking Union Medical College, 100021 Beijing, China. [4]Research Center for Translational Medicine, East Hospital, Tongji University School of Medicine, Shanghai 200120, China. [5]Department of Biochemistry and Molecular Biology, Tongji University School of Medicine, Shanghai 200120, China. [6]These authors contributed equally: Feng Wang, Yang Gao, Situ Xue. ✉e-mail: yuanjian229@hotmail.com; jiang.jdong@163.com; like1986@163.com

posttranslational modifications that affect its stability and oncogenic activity. To date, phosphorylation has been clearly shown to regulate MYC-mediated biological activities. Although the MYC oncoprotein is acetylated on multiple lysine residues by acetyltransferases[14,15], the role of site-specific acetylation in mediating the biological functions of MYC is incompletely determined.

Scavenger receptor class B member 2 (SCARB2) belongs to the class B scavenger receptor family, which includes the receptor for selective cholesteryl ester uptake, scavenger receptor class B type I (SR-BI) and CD36[16,17]. A previous study indicated that both SR-BI and CD36 were more highly expressed in human HCC tissues than in the adjacent noncancerous liver tissues[18,19]. In particular, CSCs selectively use the scavenger receptor CD36 to promote their maintenance and mediate drug resistance in glioblastoma, leukemia and breast cancer by metabolic rewiring[20,21]. SCARB2 acts as a receptor in the pathogenesis of hand, foot and mouth disease[22]. Moreover, it functions as a phospholipid receptor[23] and is involved in lysosomal cholesterol export[17]. However, the role of SCARB2 in cancer, especially HCC, remains uncharacterized.

In this work, we investigate and verify the role of SCARB2 in HCC initiation and progression. We find that SCARB2 regulates cancer stem cell-like properties of HCC by enhancing MYC activity, which plays important role in promoting HCC initiation and progression. Furthermore, we demonstrate that binding of SCARB2 with MYC enhances MYC activity by interfering with HDCA3-mediated MYC deacetylation. In addition, we evaluate the therapeutic potential of targeting SCARB2 in HCC. Taken together, this study reveals SCARB2 as a regulator of HCC and a potential therapeutic target in HCC.

## Results

### CRISPR/CAS9 library screening identified SCARB2 as a critical gene for maintaining the stem cell-like characteristics of HCC cells

CSCs have a different metabolic phenotype to that of differentiated bulk tumor cells, and these specific metabolic activities directly participate in the process of tumor transformation. To identify critical metabolic genes involved in supporting stem cell-like characteristics of HCC cells in human HCC, we performed a human CRISPR/Cas9 library screen in HCC cells. Human CRISPR/Cas9 metabolic gene knockout library which contains 2,981 human metabolic genes with 30378 guide RNAs (Supplementary Data 1) was used for screening. Firstly, HCC cells were infected with the lentiviral human CRISPR/Cas9 metabolic gene knockout library at a MOI ~ 0.3 for 2 days, and then the infected cells were selected with 4 μg/ml puromycin for 3 days to generate a mutant cell pool (Fig. 1a). After puromycin selection, we deep sequenced library sgRNAs from HCCLM3 mutant pool. Based on normalized read counts, we identified around 98% of all sgRNAs retained in the HCCLM3 mutant pool (Fig. 1b, and Supplementary Data 2). Then the mutant HCC cells were seeded in ultra-low attachment dishes to sphere-forming culture. After 15 days of culture, the tumorspheres larger than 70 μm and smaller than 40 μm were harvested by cell strainers and genomic DNA was isolated for deep sequencing (Fig. 1c). Inspection of sgRNAs that were enriched in small tumorspheres compared to large tumorspheres revealed key genes inhibiting tumorspheres formation capacity. For several gene targets, multiple sgRNAs targeting the same gene were enriched in selected small tumorspheres. Among the list of genes, SCARB2 was identified as the most enrichment gene in small tumorspheres (Fig. 1d, e). All SCARB2 targeting sgRNAs were dramatically increased in small tumorspheres (Fig. 1f), implying that SCARB2 may play an important role in maintaining stem cell-like characteristics of HCC cells.

To verify the results of the stemness screen, we validated each of the top 10 genes enriched in small tumorspheres. Most screened sgRNAs that targeted these 10 genes decreased the tumorsphere formation capacity of HCC cells (Supplementary Fig. 1a). In particular,

SCARB2 deletion (Supplementary Fig. 1b) decreased the proliferation (Supplementary Fig. 1c) and invasion (Supplementary Fig. 1d) of HCCLM3 and HepG2 cells and suppressed tumorsphere formation of HCCML3, HepG2, and primary human HCC cells (HCC[1#]) freshly isolated from patient-derived tumors (Supplementary Fig. 1e). The limiting-dilution assays in vitro showed that the frequency of tumor-initiating cells capable of forming spheres decreased after SCARB2 knockout in HCC cells (Supplementary Fig. 1f). HCC CSCs are suggested to cause drug resistance and metastasis due to their ability to self-renew and differentiate into heterogeneous lineages of cancer cells[24]. CD24, EpCAM, CD13, and CD133 have been used as liver CSCs surface markers[8,25,26]. We found that SCARB2 positive liver cancer cells co-expressed these known liver CSC markers (Supplementary Fig. 1g). In addition, the proportion of CD24, EpCAM, CD13, or CD133 positive cells was decreased after SCARB2 knockout in HCCML3 cells (Fig. 1g).

We further examined the role of SCARB2 in HCC. CD133 and CD13 have been widely used as liver CSCs surface marker, and CD133+CD13+ subpopulations exhibited enhanced stem cell-like properties than CD133-CD13- subpopulations in HCC[27]. CD133+CD13+ and CD133-CD13- subpopulations were sorted from HCCLM3, HepG2, and primary HCC cells by flow cytometry (Supplementary Fig. 1h). We found that SCARB2 was more strongly expressed in CD133+CD13+ than CD133-CD13- subpopulations (Fig. 1h). When SCARB2 was knocked down in CD133+CD13+ subpopulations, the expression of stem markers (CD24 and EpCAM) and stem transcription factors (NANOG, SOX2, and OCT4) were reduced (Supplementary Fig. 1i). SCARB2 deficiency showed strong inhibition of proliferation in CD133+CD13+ subpopulations, but had slight effect on CD133-CD13- subpopulations (Fig. 1i). Similar with previous observation, SCARB2 knockout in CD133+CD13+ subpopulations suppressed their capacity of sphere formation (Fig. 1j). These data suggested that SCARB2 plays different roles in the CD133+CD13+ subpopulations and CD133-CD13- subpopulations.

Furthermore, loss of SCARB2 decreased the median inhibitory concentration (IC$_{50}$) of sorafenib in HCC cells, which suggested that SCARB2 knockout sensitized HCC cells to sorafenib (Fig. 1k). Tumor organoids with SCARB2 deletion (Supplementary Fig. 1b) showed a higher apoptosis ratio than their control counterparts when treated with sorafenib (Fig. 1l). The proliferation and invasion of tumor organoids were decreased when SCARB2 was knocked down (Fig. 1m, n). Additionally, the proportion of CD24, EpCAM, CD13, or CD133 positive cells were reduced in SCARB2 knockout tumor organoids (Fig. 1o). These data suggested that SCARB2 may act as a critical gene for maintaining the stem cell-like characteristics of HCC cells.

We further analyzed SCARB2 abundance in a publicly available HCC patients datasets from the cancer genome atlas (TCGA) using the UALCAN platform[28] (https://ualcan.path.uab.edu/analysis.html). SCARB2 expression was enhanced in tumor compared to that in normal liver tissues (Supplementary Fig. 1j) and up-regulation of SCARB2 was observed in every stage of HCC (Supplementary Fig. 1k). We further analyzed the correlation between SCARB2, SCARB1 or CD36 expression with the known CSC markers gene in HCC using GEPIA database[29] (http://gepia.cancer-pku.cn). We found that although SCARB1 and CD36 also increased in HCC tissues compared to that in normal liver tissues (Supplementary Fig. 1j), only SCARB2 expression but not SCARB1 or CD36 was positively correlated with known CSC markers (EpCAM, CD24, CD133, or CD13) in HCC patients (Supplementary Fig. 1l). Drug resistance is an important feature of CSCs. We found that SCARB2 expression was elevated in sorafenib-resistant HCC cells compared to their parental cells (Supplementary Fig. 1m). Furthermore, we analyzed the protein level of SCARB2 in HCC specimens using immunohistochemistry staining. Compared with normal liver tissues, HCC specimens showed high expression of SCARB2 (Fig. 1p). Moreover, high expression of SCARB2 was associated with short survival times in patients with HCC (Fig. 1q). These data suggest that

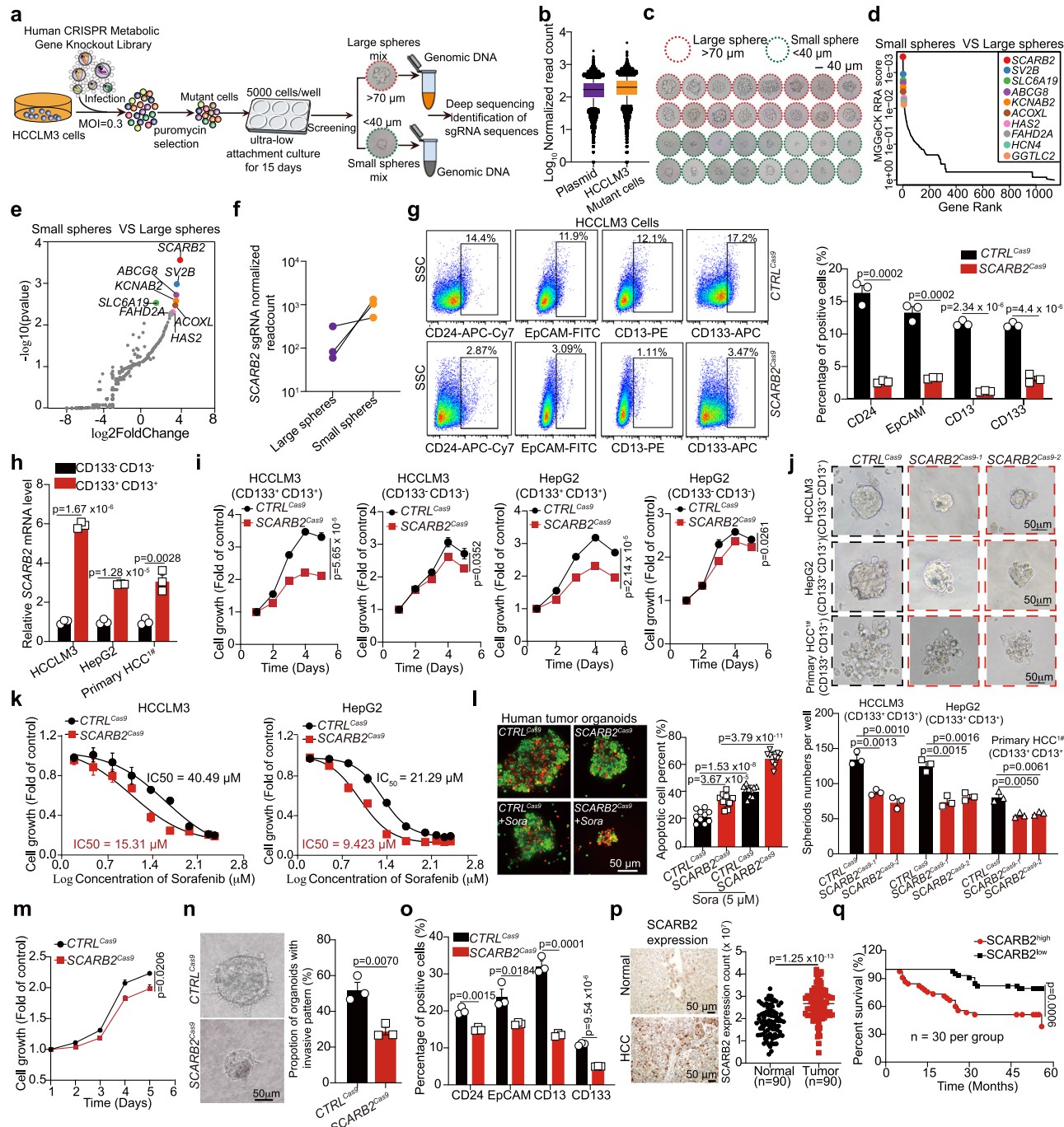

SCARB2 may perform critical pathogenic functions by supporting stem cell-like characteristics in HCC cells.

### SCARB2 deletion reduces the tumor growth and matastasis

To further investigate the role of SCARB2 in HCC, we generated mice with liver-specific conditional *Scarb2* knockout (Supplementary Fig. 2a) by crossing *Scarb2*-floxed (*Scarb2F/F*) mice and mice with hepatocyte-specific expression of Cre recombinase (*CreAlb*) (Supplementary Fig. 2b). Abrogation of SCARB2 expression was confirmed by the reduction in the SCARB2 protein level in the hepatocytes of mice with homozygous deletion of *Scarb2* (Supplementary Fig. 2c). The body and liver weights of 10-week–old *CreAlbScarb2F/F* mice were comparable to those of the control *WT* mice (Supplementary Fig. 2d). Macroscopically, *Scarb2* knockout livers appeared normal (Supplementary Fig. 2e). There was no significant difference in serum alanine aminotransferase (ALT),

aspartate aminotransferase (AST), cholesterol (CHO), glucose (GLU), and triglyceride (TG) levels for these two groups (Supplementary Fig. 2f), indicating that deletion of *Scarb2* didn't affect liver homeostasis. These *CreAlbScarb2F/F* mice were further crossed with *H11-CAG-LSL-Myc* mice to generate heterozygous *Scarb2* knockout *CreAlbScarb2F/+Myc* mice and homozygous *Scarb2* knockout *CreAlbScarb2F/FMyc* mice (Fig. 2a and Supplementary Fig. 2g). After mating *H11-CAG-LSL-Myc* mice with *CreAlb* mice, liver cancer developed spontaneously in 6- to 8-week-old *CreAlbMyc* mice. Heterozygous knockout of *Scarb2* decreased the tumor sizes and weights (Fig. 2b, c). The tumor sizes and weights were further decreased in *CreAlbScarb2F/FMyc* mice (Fig. 2b, c). Furthermore, even with heterozygous knockout of *Scarb2*, the tumor incidence rate decreased, and the survival time of *CreAlbScarb2F/+Myc* mice was extended based on Kaplan–Meier analysis (Fig. 2d, e). *CreAlbScarb2F/FMyc* mice showed a significantly lower incidence of HCC and longer survival time

**Fig. 1 | CRISPR/Cas9 library screening identified SCARB2 as a positive modulator of stem cell-like characteristics of HCC cells. a** Schematic diagram illustrating the workflow for human CRISPR/Cas9 library screening. **b** The normalized read counts of all sgRNAs in CRISPR/Cas9 library plasmid ($n = 29,957$) and HCCLM3 cells infected with human CRISPR/Cas9 library ($n = 29,875$). The center line indicates the 50th percentiles; bounds of box = 25th–75th percentiles; whiskers = 10th–90th percentiles; minima: bottom whiskers; maxima: top whiskers. **c** Representative images of tumorspheres. large spheres (>70 μm); smaller spheres (<40 μm). **d** MAGeCK analysis and RRA ranking of top enriched genes in small tumorspheres compared to large tumorspheres. **e** Ranked dot plots of top enriched genes in small tumorspheres compared to large tumorspheres. *P* value was obtained by permutation test using Benjamini-Hochberg procedure by MAGeCK. **f** The normalized read counts of sgRNAs targeting *SCARB2* between large tumorspheres and small tumorspheres. **g** Flow cytometric analysis of the proportion of CD24, EpCAM, CD13, or CD133 positive cells in *CTRL^Cas9* or *SCARB2^Cas9* HCCLM3 cells. **h** Real-time PCR analysis of *SCARB2* expression in CD133⁺ CD13⁺ and CD133⁻ CD13⁻ HCCLM3, HepG2 and primary HCC cells. **i** Cell growth curves of CD133⁺ CD13⁺and CD133⁻ CD13⁻ HCCLM3 and HepG2 cells with or without *SCARB2* deletion.

**j** Representative images of tumorspheres of indicated cells with or without *SCARB2* knockout. The number of spheres was counted. Scale bar, 50 μm. **k** Effect of *SCARB2* depletion on sorafenib sensitivity in HCCLM3 and HepG2 cells. The data are a summary of IC₅₀ values for sorafenib. **l** Representative immunofluorescence images and quantification of the viability of HCC organoids with indicated treatment. Calcein acetoxymethyl (calcein-AM) was used to mark viable cells (green) and ethidium bromide homodimer-1 to mark dead cells (red) ($n = 10$ organoids per group). Scale bar, 50 μm. **m** Relative cell viabilities of tumor organoids with or without *SCARB2* knockout. **n** Representative images and quantification of protrusive invasion of HCC organoids. Scale bar, 50 μm. **o** Flow cytometric analysis of the proportion of CD133, CD13, EpCAM or CD24 positive cells in HCC organoids with or without *SCARB2* knockout. **p** IHC staining of SCARB2 expression in human normal liver tissues and HCC specimens ($n = 90$ per group). **q** Kaplan–Meier survival curves for patients with HCC stratified by SCARB2 expression. (**g**, **h**, **i**, **j**, **k**, **m**, **n**, **o**) $n = 3$ biological repeats. Statistical significance was calculated by (**g**, **h**, **i**, **j**, **l**, **m**, **n**, **o**, **p**) two-tailed Student's *t* test; (**q**) two-sided log-rank test; Data are presented as means ± S.E.M. Source data are provided as a Source Data file.

than *Cre^AlbScarb2^F/+Myc* mice and *Cre^AlbMyc* mice (Fig. 2d, e). Consistent with these results, LDA to assess the tumor repopulation ability revealed that tumor cells from *Cre^AlbScarb2^F/FMyc* and *Cre^AlbScarb2^F/+Myc* mice exhibited reduced tumor initiation efficiency (Fig. 2f). The proportion of EpCAM, CD133, or CD24 positive cells significantly decreased in *Cre^AlbScarb2^F/FMyc* mice than that in *Cre^AlbMyc* mice, demonstrating that knockout of *Scarb2* had effect on the HCC initiating state (Fig. 2g). We then explored the effect of *Scarb2* knockout on HCC initiation in HCC mouse model induced by diethylnitrosamine (DEN) (Fig. 2h). Knockout of *Scarb2* in hepatocytes dramatically decreased the tumor sizes, liver weights, and tumor nodules in the DEN-induced hepatocarcinogenesis model (Fig. 2i-k). To confirm the inhibitory effect of *SCARB2* deletion on HCC progression, we knocked out *SCARB2* in HCCLM3 cells using lentivirus-mediated delivery of CRISPR-Cas9 system components. *SCARB2* deletion impeded the growth (Fig. 2l) and decreased the sizes and weights (Fig. 2m, n) of tumors in mice inoculated with HCCLM3 cells. Indeed, *SCARB2* deletion also decreased the number of lung metastatic nodules formed by HCCLM3 cells (Supplementary Fig. 2h, i). We further investigated the role of *SCARB2* in liver CSCs in vivo. Sphere formation is used to enrich CSCs from HCCLM3 cells[30], and these tumor spheroids were infected with *CTRL^Cas9* and *SCARB2^Cas9* virus and subcutaneously inoculated into BALB/c nude mice. *SCARB2* knockout in tumor spheroids resulted in significantly decreased tumor growth and reduced final tumor sizes and tumor weights (Fig. 2o-q and Supplementary Fig. 2j). At the endpoint of inoculation days, the tumors generated from *SCARB2* knockout tumor spheroids showed the decreased proportions of CD24, EpCAM, CD13, or CD133 positive cells (Fig. 2r). Collectively, these data suggest that *Scarb2* is essential for hepatocarcinogenesis and *Scarb2* deletion reduces the tumor growth and metastasis.

## SCARB2 promotes hepatocarcinogenesis by activating MYC

To investigate the mechanism by which SCARB2 promotes HCC pathogenesis, we analyzed the gene expression profiles of HCC cells with or without *Scarb2* deletion through RNA sequencing (RNA-seq) (Fig. 3a). Gene set enrichment analysis (GSEA) showed that HCC cells from *Cre^AlbMyc* mice exhibited high enrichment of MYC target gene signatures (Fig. 3b) among the top enriched pathways. Several MYC target gene sets were significantly and positively enriched in HCC cells from *Cre^AlbMyc* mice relative to HCC cells from *Cre^AlbScarb2^F/FMyc* mice (Fig. 3c, d). Single-cell RNA sequencing (scRNA-seq) of human primary HCC tumorspheres indicated that *SCARB2* positive cells accounted about 11.01% of total tumorspheres (Fig. 3e, f) and exhibited higher level of MYC target genes compared to *SCARB2* negative cells (Fig. 3g–i). In addition, *SCARB2* expression was positively correlated with MYC target gene score and *SCARB2* positive cells exhibited high

enrichment of MYC target gene signatures in scRNA-seq data from human primary HCC tumorspheres (Fig. 3j and Supplementary Fig. 3c). qRT-PCR and WB assays showed that *Scarb2* deletion suppressed the transcription and translation of several critical MYC target genes in mouse derived HCC cells (Supplementary Fig. 3a, b). The analysis of a publicly available HCC patient datasets from GEPIA[29] (http://gepia.cancer-pku.cn) revealed that there was positive correlation between *SCARB2* with MYC target genes (*CDK4, LDHA, GLUT1, CCNA1, PLD6*, and *EIF3A*) (Supplementary Fig. 3d). Then we explored the effect of SCARB2 on MYC transcriptional activity by using a MYC luciferase reporter plasmid. *SCARB2* depletion decreased MYC transcriptional activity in HCC cells, while *SCARB2* overexpression had the opposite effect (Fig. 3k). We next performed CUT&Tag assay in HCC cells with or without *SCARB2* deletion to investigate whether *SCARB2* affects MYC activity by mapping promoter occupancy of MYC in HCC. The results showed that *SCARB2* deletion significantly reduced the MYC occupancy in the genome and diminished the MYC binding in the promoter of MYC target genes *CDK4* and *SLC2A1* (Fig. 3l–n). We further validated the CUT&Tag result with ChIP-qPCR: *SCARB2* knockout in HCCLM3 cells or *Scarb2* knockout in mouse HCC cells in vivo reduced the binding of MYC to MYC target genes, including *CAD, CDK4, LDHA, NCL, PKM2* and *HES1* (Fig. 3o, p). The DNA binding and transcriptional activity of MYC requires its dimerization with MAX[31]. We found that *SCARB2* deficiency decreased MYC/MAX interaction and the formation of MYC/MAX heterodimer (Fig. 3q, r). Taken together, these data indicate that SCARB2 promotes hepatocarcinogenesis by enhancing MYC transcriptional activity and increasing the expression of its target genes.

## SCARB2 interacts with MYC to disrupt HDAC3-mediated MYC deacetylation

Posttranslational modifications, including phosphorylation, ubiquitylation, and acetylation, play critical roles in controlling MYC expression and activity. SCARB2 overexpression did not affect the total phosphorylation level of MYC on serine (Ser, S) and threonine (Thr, T) residues (Supplementary Fig. 4a) or the ubiquitylation of MYC (Supplementary Fig. 4b). However, *SCARB2* deletion reduced the level of MYC acetylation (Fig. 4a), and *SCARB2* overexpression had the opposite effect (Fig. 4b). We further explored the detailed mechanism by which MYC acetylation contributes to the regulation of MYC transcriptional activity by SCARB2. Quantitative mass spectrometry (MS)-based proteomics was used to assess acetylation modification of MYC in HCCLM3 cells with or without *SCARB2* overexpression. Peptides with acetylation on lysine (K) 148 and 326 of MYC were enriched in SCARB2-overexpressing HCCLM3 cells (Table 1). To further confirm the trends observed in our mass

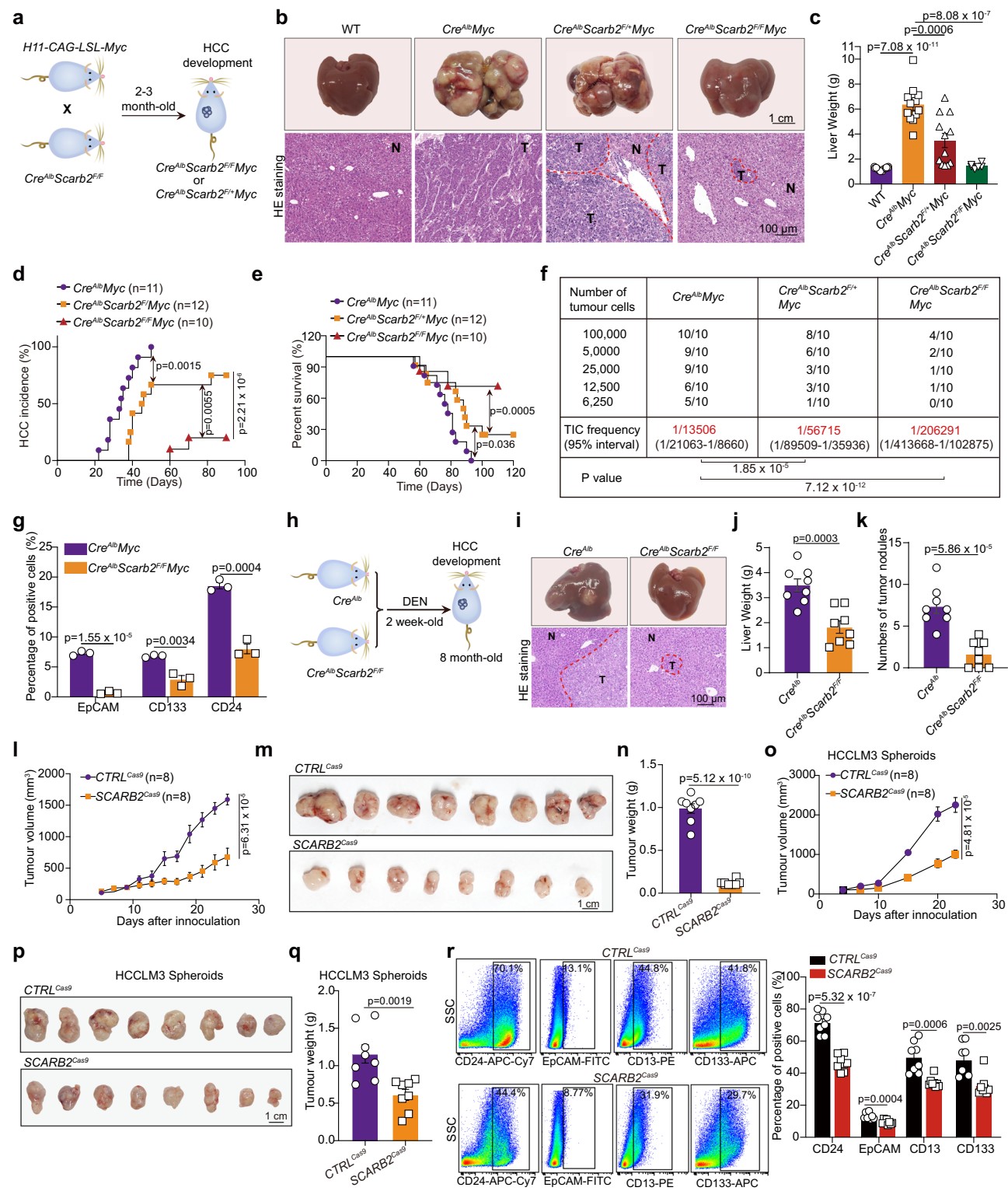

spectrometry-based analysis, we next mutated K148 and K326 to the nonacetylatable amino acid arginine (R) and quantified the acetylation level of the MYC K148R and K326R mutants in HEK 293 T cells with or without *SCARB2* overexpression. Compared to wild-type MYC, both the K148R and K326R MYC mutants exhibited reduced acetylation (Fig. 4c and Supplementary Fig. 4c). However, the K148R mutation but not the K326R mutation diminished the effect of SCARB2 overexpression on MYC acetylation (Fig. 4c, and Supplementary Fig. 4c). These data indicate that SCARB2 upregulates MYC acetylation on K148.

p300[14], HDACs[32] KAT5[33], SIRT1[34], and GCN5[35] affected MYC acetylation and deacetylation in both direct and indirect manners. We found that SCARB2 overexpression further increased MYC acetylation in p300-, KAT5-, or GCN5 silencing HCC cells (Supplementary Fig. 4d), suggesting that one or more additional acetyltransferases or deacetylases may be involved in this SCARB2-regulated process. However, the decrease in MYC acetylation mediated by the deacetylase HDAC3 but not SIRT1 was restored by overexpression of SCARB2 in HCC cells (Fig. 4d and Supplementary Fig. 4e), indicating that SCARB2 affects MYC acetylation mainly in an HDAC3-dependent manner. We next

**Fig. 2 | *SCARB2* deletion reduces the tumor growth and metastasis. a** Scheme used to establish the model of spontaneous HCC with targeted *Myc* knock-in and *Scarb2* knockout in the liver. **b** Representative photographs (top) and H&E staining (bottom) of intrahepatic tumor tissues in the indicated mice 8 weeks after birth. Scale bar, 1 cm. H&E staining Scale bar, 100 μm. **c** The liver weights of *WT* (*n* = 12 mice), *Cre^Alb^ Myc* (*n* = 12 mice), *Cre^Alb^ Scarb2^F/+^Myc* (*n* = 12 mice) or *Cre^Alb^ Scarb2^F/F^Myc* mice (*n* = 6 mice). **d** Incidence of HCC in *Cre^Alb^ Myc* (*n* = 11 mice), *Cre^Alb^ Scarb2^F/+^Myc* (*n* = 12 mice) or *Cre^Alb^ Scarb2^F/F^Myc* mice (*n* = 10 mice). **e** Kaplan–Meier survival curves for *Cre^Alb^ Myc* (*n* = 11 mice), *Cre^Alb^ Scarb2^F/+^Myc* (*n* = 12 mice) or *Cre^Alb^ Scarb2^F/F^Myc* mice (*n* = 10 mice). **f** The tumor initiation efficiency of HCC cells harvested from indicated group was evaluated by in vivo limiting dilution assay (*n* = 10 mice per group). **g** Flow cytometric analysis of the proportion of EpCAM, CD133 or CD24 positive cells in indicated group (*n* = 3 mice per group). **h** Scheme used to establish DEN-induced HCC mouse model. **i** Representative photographs (top) and

H&E staining (bottom) of intrahepatic tumor tissue in the indicated mice 8 months after birth. **j** Liver weights of the *Cre^Alb^* mice (*n* = 8 mice) and *Cre^Alb^ Scarb2^F/F^* mice (*n* = 8 mice) in DEN-induced HCC mouse model. **k** Numbers of tumor nodules in the indicated mice (*n* = 8 mice per group). **l** Effect of *SCARB2* knockout in HCCLM3 cells on tumor growth (*n* = 8 mice per group). Representative images of tumors (**m**) and tumor weights (**n**) in the indicated group (*n* = 8 mice per group). **o**–**q** Effects of *SCARB2* knockout in HCCLM3 spheroids on tumor growth, tumor sizes and tumor weights (*n* = 8 mice per group). **r** Flow cytometric analysis of the proportion of CD24, EpCAM, CD13, or CD133 positive cells in tumors generated from HCCLM3 spheroids with or without *SCARB2* knockout (*n* = 8 mice per group). Statistical significance was calculated by (**c, g, j, k, l, n, o, q, r**) two tailed Student's *t* test; (**d, e**) two-sided log-rank test; (**f**) one-sided extreme limiting dilution analysis. Data are presented as means ± S.E.M. Source data are provided as a Source Data file.

applied CRISPR/Cas9 gene editing to generate *MYC^K148R^* mutant HCCLM3 cell lines with or without *SCARB2* deletion (Fig. 4e). The *K148R* mutation not only reduced MYC acetylation but also diminished the effect of HDAC3 overexpression on MYC deacetylation (Fig. 4f). Moreover, the *K148R* mutation decreased the transcriptional activity of MYC compared to that of *MYC^WT^*; However, *SCARB2* deletion only reduced MYC transcriptional activity in *MYC^WT^* but not *MYC^K148R^* HCCLM3 cells (Fig. 4g). The ChIP-qPCR result showed that *K148R* mutation of MYC reduced the binding of MYC to its target genes, including *CAD, CDK4, LDHA, NCL, PKM2* and *HES1* gene promoters compared to *MYC* wildtype (Fig. 4h).Given that K148, located in the CT domain of MYC, mediates the interaction of MYC with several cofactors, including GCN5, KAT5, and BRD4[36], we examined whether acetylated K148 of MYC is the binding site between MYC and these cofactors. Endogenous coimmunoprecipitation (Co-IP) indicated that the *K148R* mutation impeded the ability of MYC to interact with BRD4, KAT5, and GCN5 (Fig. 4i). Moreover, we also investigated the consequence of *MYC^K148R^* acetylation on the proliferation and tumorsphere formation of HCC cells. HCCLM3 cells with *MYC^K148R^* mutation exhibited a remarkable decrease in the proliferation and the sphere-forming ability compared to that of *MYC^WT^* HCCLM3 cells (Fig. 4j, k). However, *SCARB2* deletion couldn't further reduce the proliferation or the sphere-forming ability in *MYC^K148R^* mutant HCCLM3 cells (Fig. 4j, k). Taken together, these data suggest that K148 acetylation enhanced by SCARB2 is critical for MYC transcriptional activity by providing the docking site for critical cofactors with MYC (Fig. 4l).

We next explored the molecular mechanism of the SCARB2 protein in HDAC3-mediated MYC deacetylation. To identify the potential binding partner of SCARB2, we performed Quantitative mass spectrometry (MS)-based proteomics to analysis of SCARB2 immunoprecipitates. MYC ranked among the top 10 proteins that coimmunoprecipitated with SCARB2 (Fig. 5a), and HDCA3 was also one of potential interacting partners of SCARB2 (Supplementary Fig 4f). MYC was identified to coimmunoprecipitate with SCARB2 and HDAC3 (Fig. 5b and Supplementary Fig. 4g, h), and the interaction of MYC and SCARB2 located in the cytoplasm and nucleus but not in the cytomembrane of HCCLM3 cells (Fig. 5c). In addition, the MYC/SCARB2 colocalization was observed in HepG2 cells (Fig. 5d). Deletion of SCARB2 increased but overexpression of SCARB2 reduced the interaction and colocalization of HDAC3 with MYC (Fig. 5e, f and Supplementary Fig. 4i). Then we mapped the interaction domain of MYC with SCARB2 and HDAC3. The CT element and N-terminal domain (NTD) of MYC were responsible for the interaction of MYC with SCARB2 and HDAC3 (Fig. 5g), suggesting that SCARB2 shared the same interaction domain of MYC with HDAC3. Moreover, using SCARB2 mutants, we identified domains involved in the interaction between SCARB2 and MYC. Deletion of the C-terminal transmembrane domain (CTM), N-terminal transmembrane domain (NTM) or cytoplasmic domain (CD) did not affect the interaction of SCARB2 and MYC, but luminal domain (LD) deletion significantly decreased the interaction of

SCARB2 and MYC, indicating that the LD domain is the primary contributor to this interaction (Fig. 5h).

We then evaluated the effect of HDAC3 inhibition on the binding of MYC to chromatin. The ChIP-qPCR result revealed that HCCLM3 cells with RGFP966 treatment[37] increased the binding of MYC to MYC target genes (Fig. 5i). Furthermore, we found that *HDAC3* but not *HDAC3 R26SP* mutation (an inactivation mutation in HDCA3)[38] overexpression decreased the proliferation and the sphere-forming ability in *SCARB2*-deleted HCC cells (Fig. 5j, k). Taken together, these data indicated that SCARB2 competes with HDAC3 to bind with MYC, which in turn elevates MYC acetylation and promotes proliferation and colony-formation of HCC cells.

## Polymyxin B (PMB) binds with SCARB2 to counteract HCC progression

Repurposing of already-approved drugs for other indications may be an alternative strategy for drug development. We next performed virtual screening and binding assays using AutoDock Vina, a docking program that computationally examines the binding energy of a compound to its target, to identify small molecules that target the luminal domain (LD) of SCARB2 (Fig. 6a). We ranked 1317 Food and Drug Administration (FDA)-approved drugs according to their simulated binding energy. We then selected the top 10 compounds with a docking energy ≤−10.0 kcal/mol (Fig. 6b) and examined the binding abilities of these 10 candidate SCARB2 inhibitors using surface plasmon resonance (SPR) analysis. Four drugs–Goserelin acetate, Tannic acid, polymyxin B (PMB), and Alarelin exhibited binding to SCARB2 (Supplementary Fig. 5a and Fig. 6c). Among these drugs, PMB displayed a high binding affinity for SCARB2 ($K_D$ = 0.231 μM; Fig. 6c). Specifically, prediction of the interacting residues of SCARB2 with PMB showed several hydrogen bonds between the LD domain of the SCARB2 protein and PMB (Fig. 6d). PMB decreased the binding affinity between SCARB2 and MYC in vitro (Fig. 6e). High-resolution structured illumination microscopy (SIM) showed the SCARB2-MYC colocalization in HCC cells treated with vehicle (CON), whereas PMB inhibited the colocalization and interaction of SCARB2 with MYC (Fig. 6f, g) and rescued the binding of MYC with HDAC3 (Fig. 6h). Furthermore, PMB reduced MYC acetylation, binding of MYC to chromatin and MYC transcriptional activity in HCC cells (Fig. 6i, j and Supplementary Fig 5b). Gene set enrichment analysis (GSEA) showed that MYC target gene sets were highly enriched in CON-treated HCC cells compared to PMB-treated HCC cells (Supplementary Fig. 5c).

Similar to SCARB2 deletion, PMB treatment suppressed the proliferation of several HCC cell lines, and the combination of PMB with sorafenib showed a synergistic effect on HCC cells (Supplementary Fig. 5d). PMB treatment decreased the tumorsphere formation capacity and led HCCLM3 cells hypersensitize to sorafenib (Fig. 6k). In human HCC tumor organoids, sorafenib treatment induced apoptosis, and this effect was further enhanced when PMB was combined with sorafenib (Fig. 6l). In addition, the frequency of tumor-initiating cells decreased

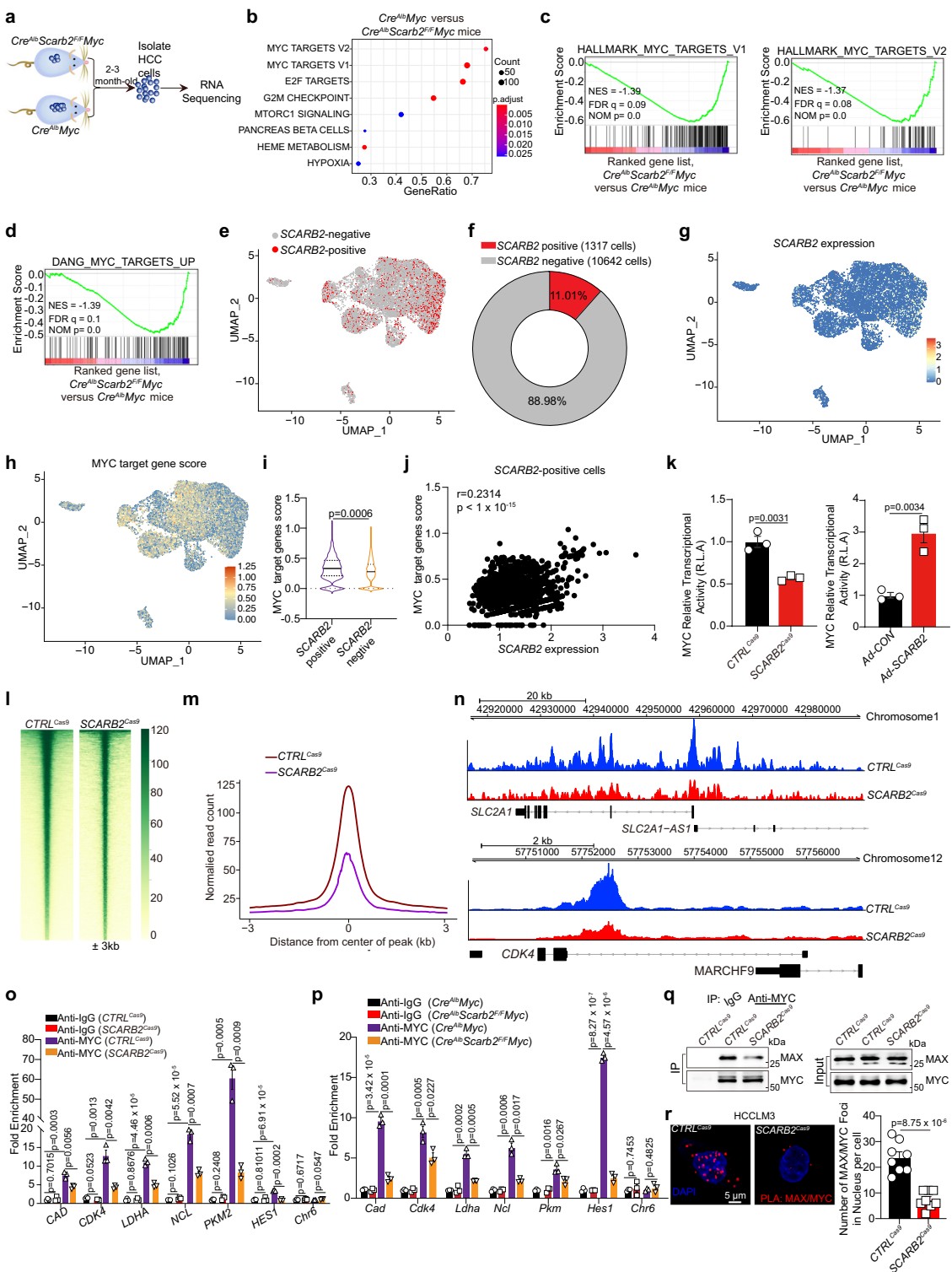

in HepG2 or human primary HCC cells treated with PMB (Fig. 6m). And the proportion of CD133, CD13, EpCAM or CD24 positive cells were decreased in human primary HCC cells with PMB treatment (Supplementary Fig. 5e). However, PMB only reduced the proliferation and sphere-forming ability of $MYC^{WT}$ but not $MYC^{K148R}$ HCCLM3 cells (Fig. 6n, o), suggesting the $MYC^{K148R}$ mutation is resistant to PMB. We further used $MYC$ or $SCARB2$ depleted HCC cells to examine the effects of PMB on cell growth and colony formation ability. $MYC$ or $SCARB2$ depletion diminished the anti-proliferation and anti- colony formation ability of PMB treatment (Fig. 6p, q), suggesting the specificity of PMB

to SCARB2/MYC cascade. Overall, these data indicate that PMB disrupts the SCARB2-MYC interaction and subsequently decreases MYC acetylation and activity to counteract HCC progression.

We further examined the antitumor effect of PMB in cell line-derived xenograft (CDX) models. PMB treatment showed no obvious side effects in the HCC mouse models, as demonstrated by the no significant difference in body/liver weights, serum ALT, AST, CHO, GLU, or TG levels of HCC mice with or without PMB treatment (Supplementary Fig. 6a-c). PMB suppressed the growth of HepG2, HCCLM3 and Hepa1-6 tumors (Fig. 7a, b), and the combination of PMB and

**Fig. 3 | *SCARB2* deletion inhibits MYC transcriptional activity. a** Approaches for RNA-seq of HCC cells in the indicated groups. **b** The top Hallmark gene sets enriched in HCC cells from *Cre^Alb^Myc* mice compared to *Cre^Alb^Scarb2^F/F^Myc* mice. *P* value was determined by one-sided permutation test. Statistical adjustments were made for multiple comparisons. **c, d** GSEA showing the enrichment of MYC target genes in *Cre^Alb^Scarb2^F/F^Myc* vs *Cre^Alb^Myc* groups. *P* value was determined by one-sided permutation test. Statistical adjustments were made for multiple comparisons. **e** Visualization of *SCARB2* positive and negative cells in tumorspheres by UMAP. **f** The proportion of *SCARB2* positive cells in tumorspheres was analyzed by scRNA-seq. **g, h** Visualization of *SCARB2* and MYC target genes expression in tumorspheres by UMAP. **i** Violin plot showing expression levels of MYC target gene score in *SCARB2* positive and negative cells. The tips of the violin plot represent minima and maxima, and the width of violin plot shows the frequency distribution of data. **j** Correlation expression of MYC target genes and *SCARB2* expression in *SCARB2* positive cells. Each data point represents the value from an individual cell. **k** Effects of *SCARB2* deletion or overexpression on the transcriptional activity of MYC.

**l** Heatmap showing occupancy of genome-wide MYC peaks in *CTRL^Cas9^* and *SCARB2^Cas9^* HCCLM3 cells in a ± 3 kb window surrounding the TSS. **m** Metagene plots of global MYC occupancy in gene bodies in *CTRL^Cas9^* and *SCARB2^Cas9^* HCCLM3 cells. **n** CUT & tag tracks showing the binding of MYC to *CDK4* and *SLC2A1* in *CTRL^Cas9^* and *SCARB2^Cas9^* HCCLM3 cells. **o, p** *CTRL^Cas9^* and *SCARB2^Cas9^* HCCLM3 cells (**o**) or HCC cells from *Cre^Alb^Myc* and *Cre^Alb^Scarb2^F/F^Myc* mice (**p**) were analyzed by ChIP with MYC or IgG antibody. ChIP'd DNA was quantified using qPCR for MYC or IgG binding to MYC target genes promoters. **q** Effect of *SCARB2* knockout on the interaction MYC with MAX. Extracts of *CTRL^Cas9^* and *SCARB2^Cas9^* HCCLM3 cells were IP anti-MYC Ab and blotted with an anti-MAX Ab. **r** Colocalization of MYC and MAX in *CTRL^Cas9^* and *SCARB2^Cas9^* HCCLM3 cells was detected by the Duolink PLA assay. Data are representative images of MYC/MAX foci (left) and quantification of the number of fluorescent foci (*n* = 8 cells per group). Scale bar, 5 μm. **k, o, p, q, r** *n* = 3 biological repeats. Statistical significance was calculated by (**i, k, o, p, r**) two tailed Student's *t* test; (**j**) two-sided Pearson's correlation test; Data are presented as means ± S.E.M. Source data are provided as a Source Data file.

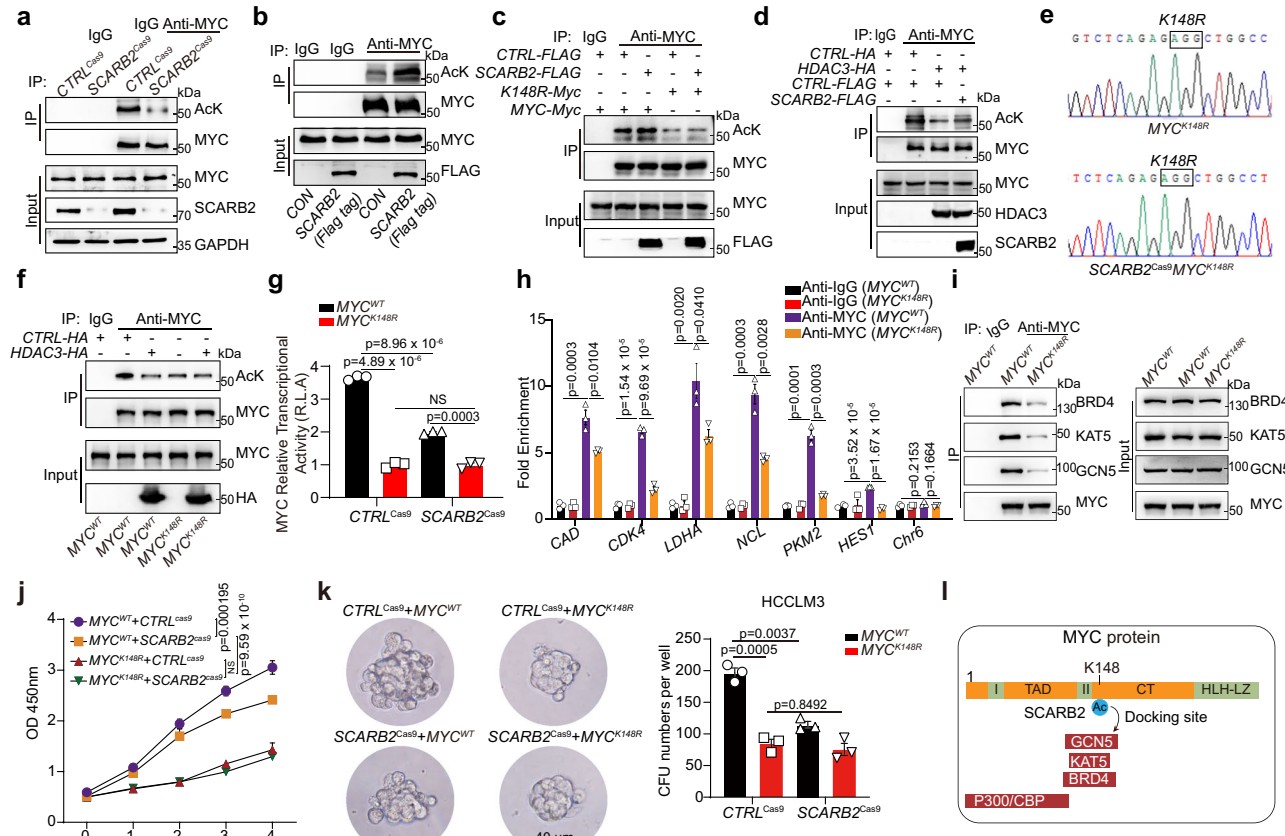

**Fig. 4 | SCARB2 disrupt HDAC3-mediated MYC deacetylation through MYC K148 site. a** The effects of *SCARB2* deletion on the acetylation of MYC were assessed by Co-IP. **b** The effects of *SCARB2* overexpression on the acetylation of MYC were assessed by Co-IP. **c** Effects of SCARB2 on the acetylation of *MYC K148R* mutants. HEK 293 T cells were transfected with the indicated plasmids for 24 h. Cell extracts were IP with an anti-MYC Ab. Acetylated MYC was detected by immunoblotting. **d** Effect of *SCARB2* overexpression on HDAC3-mediated deacetylation. HCCLM3 cells were transfected with the indicated plasmids for 24 h. Cell extracts were IP with an anti-MYC Ab. Acetylated MYC was detected by immunoblotting. **e** Sequencing verification of the codon replacement by CRISPR-Cas9 resulting in *MYC K148R*. **f** Effect of *HDAC3* overexpression on acetylation of MYC in *MYC^WT^* and *MYC^K148R^* HCCLM3 cells. **g** Effects of the *MYC K148R* mutation on MYC transcriptional activity in HCCLM3 cells with or without *SCARB2* depletion. Data are means ± S.E.M

of 3 independent experiments. **h** *MYC^WT^* and *MYC^K148R^* HCCLM3 cells were analyzed by ChIP with MYC or IgG antibody. ChIP'd DNA was quantified using qPCR for MYC or IgG binding to MYC target genes promoters, *CAD, CDK4, LDHA, NCL, PKM2, HES1*, or *Chr6* (negative control). **i** Effects of the K148R mutation on the interaction of MYC with BRD4, KAT5, and GCN5. **j** Effect of the *MYC^K148R^* mutation on the proliferation of HCCLM3 cells with or without *SCARB2* depletion. **k** Effects of the *MYC^K148R^* mutation on the sphere-forming ability of HCCLM3 cells with or without *SCARB2* depletion. **l** Schematic showing the role of MYC K148 acetylation in providing a potential docking site for binding with GCN5, KAT5, and BRD4.

**a, b, c, d, f, g, h, i, j, k** *n* = 3 biological repeats. Statistical significance was calculated by (**g, h, j, k**) two tailed Student's *t* test. Data are presented as means ± S.E.M. Source data are provided as a Source Data file.

**Table 1 | Summary of mass spectrometry (MS) analysis of acetylation site of MYC in SCARB2-overexpressing HCCLM3 cells**

| Positions within proteins | Protein | Score diff | Acetyl (K) Probabilities | Acetyl (K) Score diffs |
|---|---|---|---|---|
| 148 | NP_002458 | 89.2306 | LVSEK(1)LASYQAAR | LVSEK(89.23)LASYQAAR |
| 326 | NP_002458 | 50.4729 | RVK(1)LDSVR | RVK(50.47)LDSVR |

sorafenib showed a marked inhibitory effect on HepG2, HCCLM3 and Hepa1-6 tumor growth (Fig. 7a, b). In addition, PMB reduced the TIC frequency from 1 in ~121 cells in tumors from CON-treated HepG2 tumor-bearing mice to 1 in ~1474 cells in tumors from PMB-treated mice (Fig. 7c, upper). Similarly, PMB treatment reduced the capacity of HCCLM3 cells to form secondary tumors (Fig. 7c, bottom). Moreover, sorafenib reduced the TIC frequency in mice bearing HepG2 and HCCLM3 CDXs by ~6-fold, and these effects were more significant in mice treated with the combination of PMB and sorafenib (Fig. 7c). Mechanistically, PMB reduced the acetylation of MYC through disrupting the interaction between SCARB2 and MYC in HepG2 and HCCLM3 tumors (Supplementary Fig. 6d, e). In an established HCC patient-derived xenograft (PDX) model with positive expression of MYC and SCARB2 (Fig. 7d, e), we observed that PMB suppressed the tumor growth and the combination of PMB and sorafenib further decreased the growth and weights of PDXs in mice (Fig. 7f–h). Especially, PMB decreased ALDH$^+$ subpopulation ratio of PDX tumors with or without sorafenib treatment (Fig. 7i). These data indicate that SCARB2 plays critical role in maintaining the initiation and progression of HCC and targeting SCARB2-MYC interaction may serve as a therapeutic strategy for HCC by reducing the proliferation of HCC cells and inhibiting the stem cell-like characteristics of HCC cells and encourage immediate clinical translation of PMB-based therapeutics for HCC treatment (Fig. 7j).

## Discussion

Although targeting CSCs has been considered a promising strategy for the treatment of HCC by suppressing tumor initiation, metastasis, and chemotherapeutic resistance[3,4], relatively little is known regarding the therapeutic targets that control stem cell-like characteristics of HCC cells and contribute to HCC occurrence and progression. Therefore, deeply understanding the mechanism of maintaining stem cell-like characteristics of HCC cells will contribute to the development of targeting CSCs for HCC treatment. In this study, using CRISPR/Cas9 metabolic gene knockout library screening, we identified *SCARB2* as one of critical genes in maintaining the stem cell-like characteristics of HCC cells. Functionally, we validated the effect of SCARB2 on the proliferation, invasion and tumorspheres formation by *SCARB2* knockout in HCC cell lines and primary HCC cells. Using transgenic mice with *Scarb2* deletion in hepatocytes, we found that SCARB2 participated in the hepatocarcinogenesis. *SCARB2* knockout suppressed the proliferation, invasion and tumor initiation efficiency of HCC cells by reducing MYC transcriptional activity and acetylation. Therefore, our study thus provides insights into the function and mechanism of maintaining stem cell-like characteristics of HCC cells and also discovers one target for targeted therapies in HCC.

Deficiency in SCARB2 is related to both Gaucher disease (GD) and Parkinson's disease (PD), which are both neurodegenerative-related diseases[39,40]. SCARB2, which is involved in lysosomal cholesterol export, plays important role in lipidoses[23]. However, the relationship between SCARB2 and cancer has not yet been reported or validated. Our data suggest that SCARB2 expression was enhanced in human HCC samples compared with normal liver tissues, and *SCARB2* expression is positively correlated with known CSC markers (*CD133*, *EpCAM* or *CD24*) in HCC patients, which was consistent with the findings that another scavenger receptor CD36 was also reported to mainly enriched in CSCs and positively correlated with previously reported

CSC markers CD133 and integrin alpha 6 expression in Glioblastoma[41]. Mechanistically, SCARB2 promotes hepatocarcinogenesis and progression by regulating MYC transcriptional activity. Our RNA-seq data revealed that *SCARB2* plays important role in MTORC1 signaling pathway and HEME metabolism pathway activation. As a metabolism-related gene, whether *SCARB2* participated in the metabolism of HCC cells deserved to be further explored experimentally. Although we identified SCARB2 positive cells account for about 10% of the heterogeneous primary HCC cells, whether SCARB2 positive cells in HCC are tumor initiating cells and drive the expansion of tumor cells with stem cell characteristics needed to be further verified.

Recently, the role of MYC acetylation and deacetylation has attracted much attention due to its effect on the regulation of MYC expression and/or function through transcriptional and post-transcriptional mechanisms[15,33,42]. Previous studies have established a link between MYC and HDAC3[32], K143 and K157 of MYC are the sites of HDAC3-mediated deacetylation, and TRAF6 regulates MYC stability through HDAC3-mediated K143 and K157 deacetylation[43]. However, in this study, we found that SCARB2 supports HCC stem cell-like features by interacting with MYC, which subsequently impairs HDAC3-mediated MYC deacetylation at K148 and thereby enhancing MYC transcriptional activity. Our results revealed that K148 is an additional HDAC3-mediated deacetylation site in MYC K148 acetylation and deacetylation affect MYC transcriptional activity mainly by interfering with the recruitment of cofactors to MYC, but not influencing MYC stability. However, our study does not address the mechanistic basis for increased MYC function upon acetylation of K148. Muto et al. also reported that acetylation of K148 increased MYC activity without affecting its protein abundance in myeloid malignancies[44]. We performed the preliminary exploration of the potential mechanism by which K148 acetylation affected MYC binding to chromatin. The K148R mutation indeed decreased its interaction with MAX and the formation of MYC/MAX heterodimer. We speculated that K148 acetylation of MYC possibly changed the tertiary or spatial structure of MYC protein, making it easier to form the heterodimer of MYC/MAX. This point needs to be confirmed by the co-crystallization of full-length MYC/MAX with E-box complex.

The MYC protein interactome impacts many of the biological functions related to MYC, offering several opportunities for therapeutic intervention[45,46]. Our previous study showed that an alpha-helical peptide interfering with the protein-protein interaction (PPI) of MYC and TRIB3 displayed anti-lymphoma efficacy both in vitro and in vivo[47]. These results are encouraging and validate the quest to target MYC PPIs to diminish its oncogenic function. In this study, we verified that the MYC-SCARB2 PPI promotes the oncogenic function of MYC by interfering with HDAC3-mediated deacetylation of MYC. More importantly, we performed a screen and identified a SCARB2 binding molecule, PMB, which is a clinically available polymyxin antibiotic that has re-emerged clinically to treat infections caused by multidrug- or extensively drug-resistant gram-negative bacteria. To date, few studies have reported that PMB has anti-tumor effects in the clinic or on the bench. We identified PMB as a SCARB2 binding molecule that interferes with the PPI of SCARB2 and MYC, thereby reducing MYC acetylation and transcriptional activity. PMB treatment alone or in combination with sorafenib suppressed HCC progression in several mouse models. In our study, we found that *MYC* or *SCARB2* depletion diminished the anti-proliferation and anti- colony formation ability of

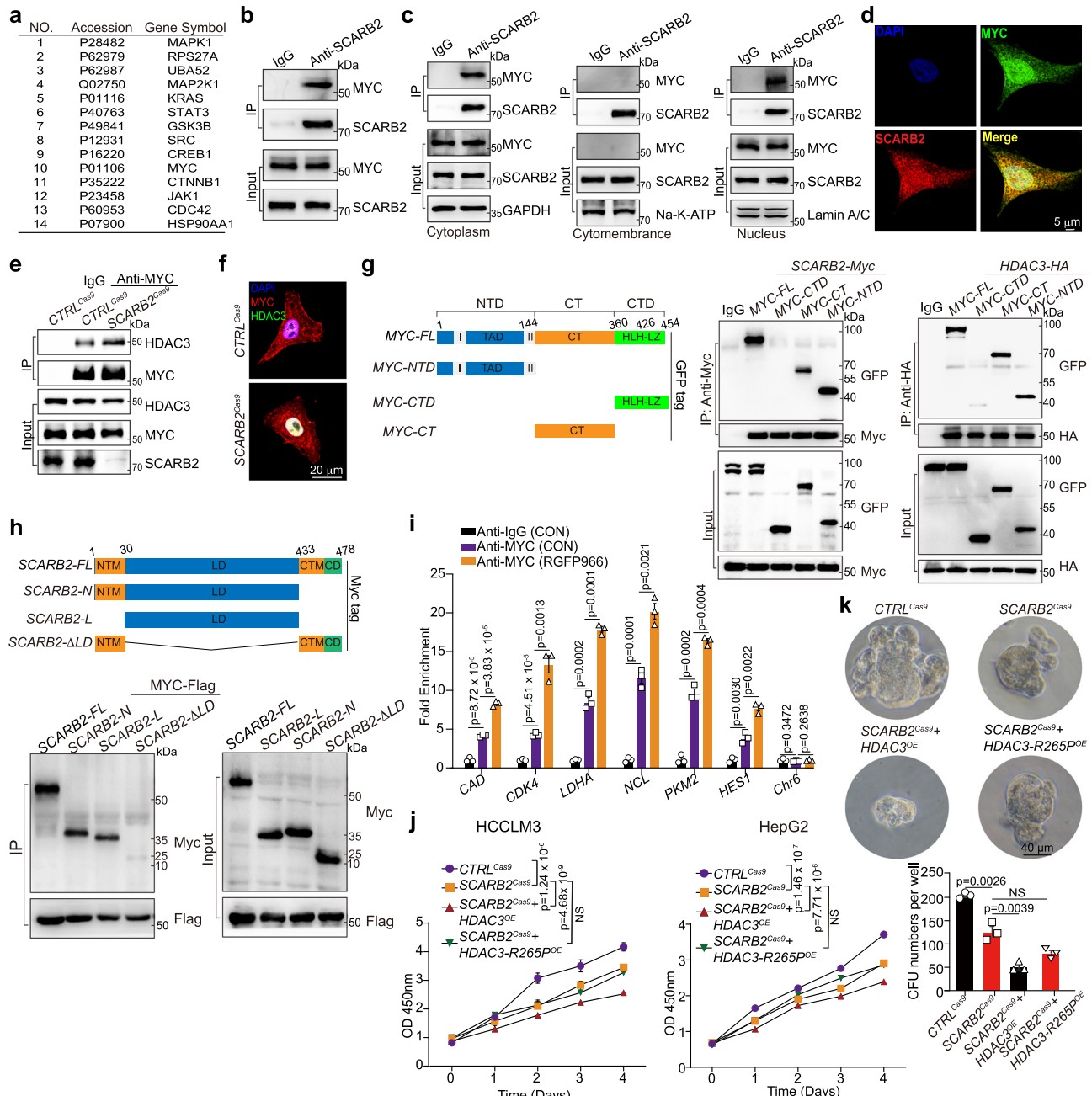

**Fig. 5 | SCARB2 interacts with MYC to disrupt HDAC3-mediated MYC deacetylation. a** MS analyzed the interaction proteins of SCARB2 in HCCLM3 cells. **b** The interaction between MYC and SCARB2 in HCCLM3 cells was evaluated by Co-IP assays. **c** CO-IP analysis of SCARB2 and MYC interaction in the cytoplasm, cytomembrane and nucleus of HCCLM3 cells. **d** Co-localization of MYC/SCARB2 was detected in HepG2 cells with immunostaining. Scale bar, 5 μm. **e** The interaction between MYC and HDAC3 in *CTRL*^Cas9 and *SCARB2*^Cas9 HCCLM3 cells was evaluated by Co-IP assays. **f** Co-localization of MYC and HDAC3 was detected in *CTRL*^Cas9 and *SCARB2*^Cas9 HepG2 cells by immunostaining. Scale bar, 20 μm. **g** Mapping of MYC regions binding to SCARB2 and HDAC3. Left: deletion mutants of MYC. Right: HEK 293 T cells were cotransfected with the indicated constructs of MYC (GFP tag) and SCARB2 (Myc tag) or HDAC3 (HA tag). Cell extracts were IP with an anti-Myc Ab or anti-HA Ab. **h** Mapping of SCARB2 regions binding to MYC. HEK 293 T cells were cotransfected with the indicated constructs of SCARB2 (Myc-tagged) and MYC (Flag-tagged). Cell extracts were IP with an anti-Flag. **i** HCCLM3 cells treated with or without HDAC3 inhibitor (RGFP966 5μM) were analyzed by ChIP with MYC or IgG antibody. ChIP'd DNA was quantified using qPCR for MYC or IgG binding to MYC target genes promoters, *CAD*, *CDK4*, *LDHA*, *NCL*, *PKM2*, *HES1*, or *Chr6* (negative control). **j** Relative cell viabilities of HCCLM3 *SCARB2*^cas9 cells or HepG2 *SCARB2*^cas9 cells with overexpression of the indicated genes for the indicated times. **k** Sphere-forming capacity of HCCLM3 *SCARB2*^cas9 cells with overexpression of *HDAC3* or *HDAC3 R265P* mutation. **b**, **c**, **d**, **e**, **f**, **g**, **h**, **i**, **j**, **k** *n* = 3 biological repeats. Statistical significance was calculated by (**i**, **j**, **k**) two tailed Student's *t* test. Data are presented as means ± S.E.M. Source data are provided as a Source Data file.

PMB treatment, suggesting that Polymyxin B has little off-target effects in HCC cells in vitro. However, it is hard to say there is not any off-target effects of Polymyxin B in vivo. The systemic treatment of Polymyxin B killed bacteria[48], induced necrosis of macrophages[49], and decreased the regulatory T cells (Treg) population[50], which may contribute to the anti-tumor activities of Polymyxin B in vivo. In summary, our study indicates that SCARB2, particularly in the context of the SCARB2-MYC PPI, is a potential therapeutic target for HCC. Furthermore, our results encourage immediate clinical translation of PMB-based therapeutics for HCC treatment.

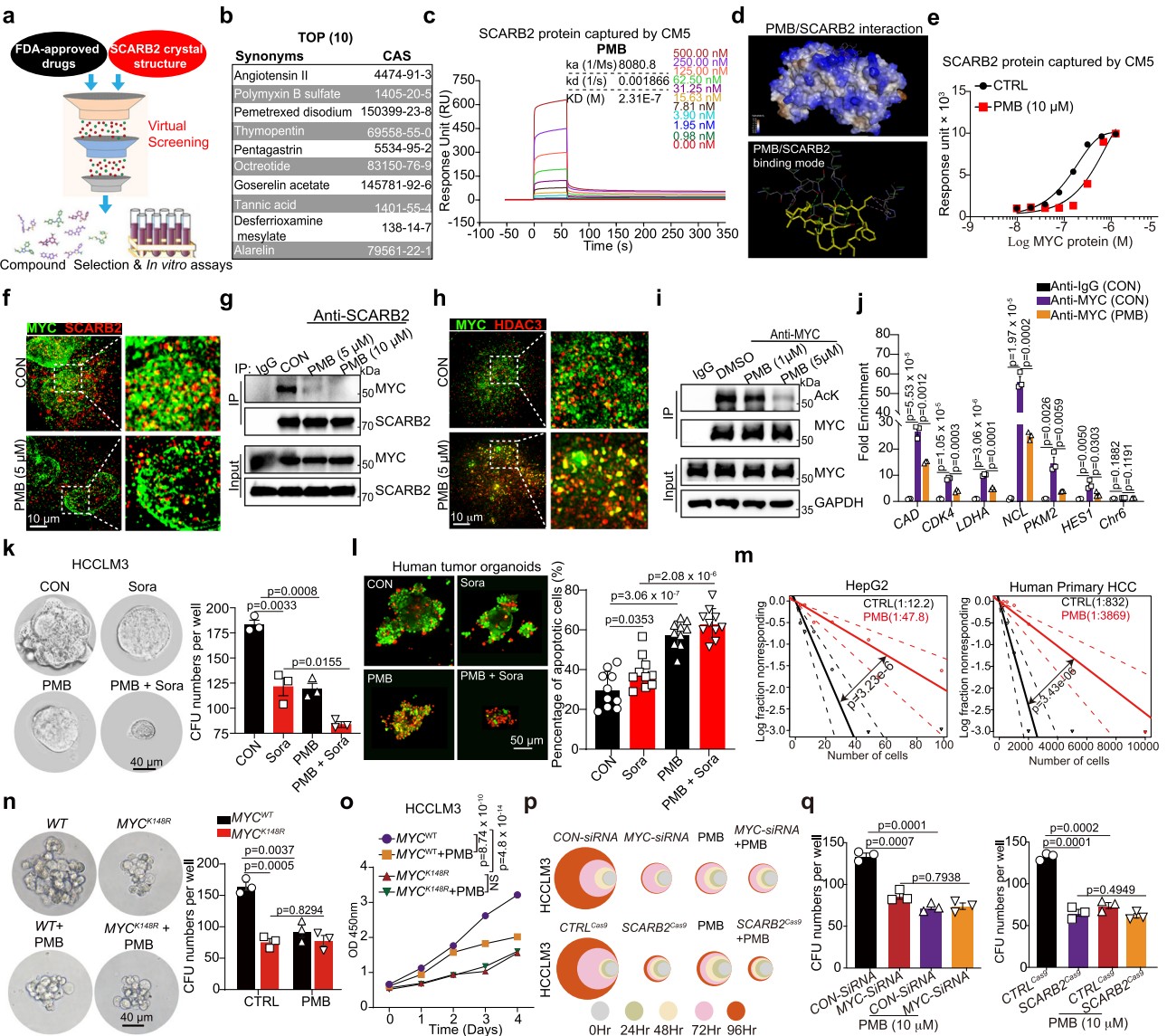

**Fig. 6 | PMB binds with SCARB2 to suppress HCC by decreasing HDAC3-mediated MYC acetylation and MYC transcriptional activity. a–b** Virtual screening of FDA-approved drugs to identify small molecules binding with SCARB2 (**a**), top 10 hits are listed (**b**). **c** The kinetics of the SCARB2-PMB interaction were determined by surface plasmon resonance (SPR) analysis. **d** The highest scoring docking model of the SCARB2 and PMB complex is shown. Top: surface of the PMB-SCARB2 complex. Bottom: 3D structure of the PMB (yellow)-SCARB2 complex. **e** The kinetics of the SCARB2-MYC interaction with or without PMB were determined by SPR. **f** Structured illumination microscopic (SIM) images of vehicle- or PMB- treated HCCLM3 cells (1 h) stained for MYC and SCARB2. Scale bar, 10 μm. **g** The effect of PMB on the interaction of MYC and SCARB2 was evaluated by Co-IP assays. Extracts of DMSO and PMB-treated HCCLM3 cells were IP with an anti-SCARB2 Ab. **h** Representative images of MYC/HDAC3 colocalization foci in HCCLM3 cells before and after PMB treatment. Scale bar, 10 μm. **i** Effect of PMB on MYC acetylation. Extracts of DMSO and PMB-treated HCCLM3 cells were IP with an anti-MYC Ab. Acetylated MYC was detected by immunoblotting. **j** HCCLM3 cells

treated with or without PMB were analyzed by ChIP with MYC or IgG antibody. ChIP'd DNA was quantified using qPCR for MYC or IgG binding to MYC target genes promoters. **k** Representative images and quantification of tumorspheres formed by HCCLM3 cells with indicated treatment. **l** Representative images and quantification of the viability of human-derived HCC organoids in 3D culture following the indicated treatment (10 organoids per group). **m** The frequency of tumor-initiating cells of HCC cells with or without PMB treatment were detected by in vitro limiting-dilution assays ($n = 10$ per group). **n–o** Effects of the $MYC^{K148R}$ mutation on the sphere-forming ability (**n**) and proliferation (**o**) of HCCLM3 cells with or without PMB treatment. **p–q** Relative cell viabilities (**p**) and tumorsphere formation (**q**) of MYC- or SCARB2-depleted HCCLM3 cells with or without PMB treatment. **f, g, h, i, j, k, n, o, p, q** $n = 3$ biological repeats. Statistical significance was calculated by (**j, k, l, n, o, q**) two tailed Student's $t$ test; (**m**) one-sided extreme limiting dilution analysis. Data are presented as means ± S.E.M. Source data are provided as a Source Data file.

## Methods
### Study approval
Primary human liver cancer and adjacent tissues were obtained from HCC patients at the Cancer Hospital, Chinese Academy of Medical Sciences. Informed consent was obtained from all participants, in accordance with the Declaration of Helsinki. The study was approved by the Ethics Committee of the Cancer Hospital, Chinese Academy of

Medical Sciences. All participants provided written informed consent to publish information that identifies individuals. Our study is compliant with the 'Guidance of the Ministry of Science and Technology (MOST) for the Review and Approval of Human Genetic Resources', which requires formal approval for the export of human genetic material or data from China. Patient-related information is provided in Supplementary Table 1.

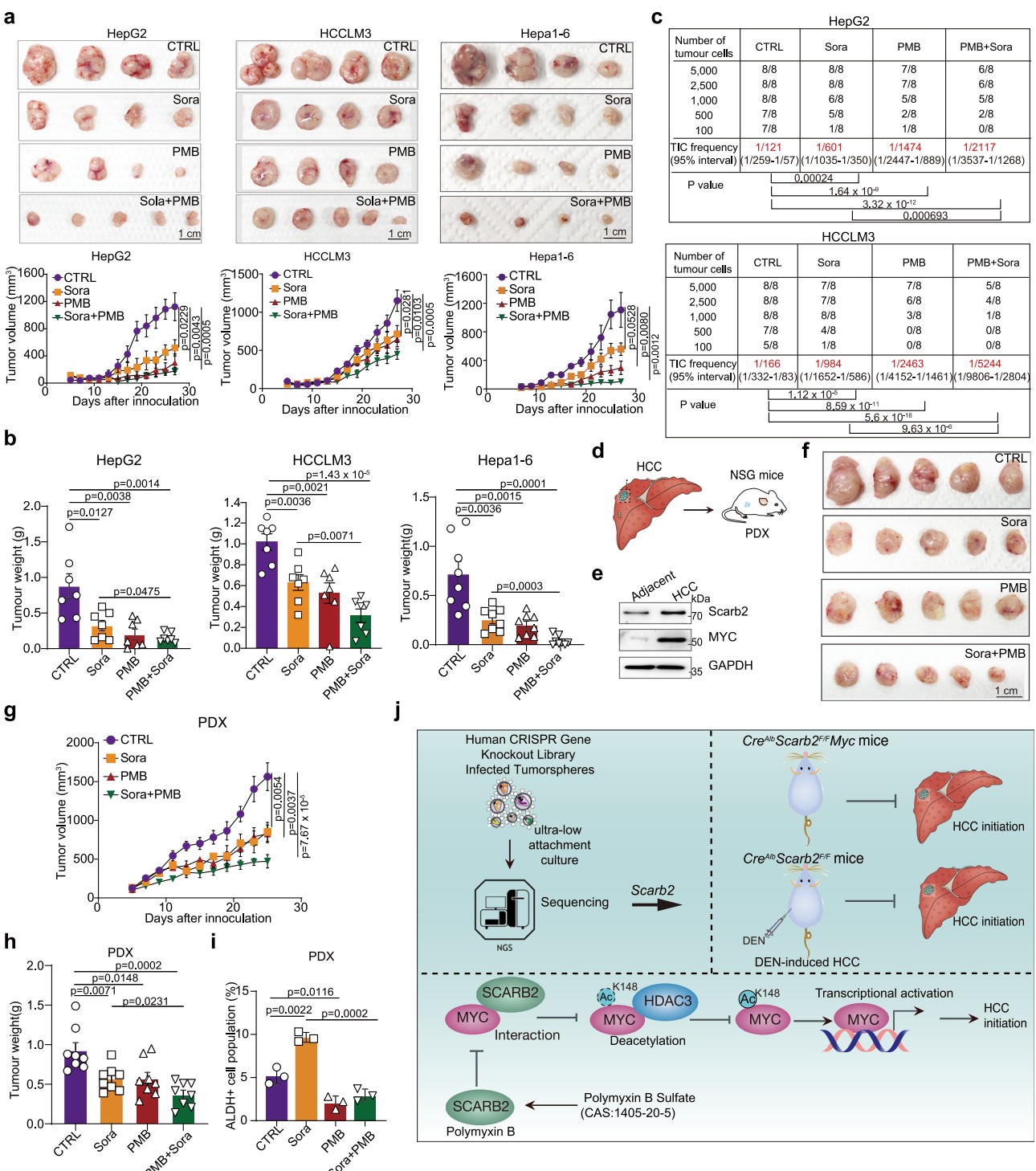

**Fig. 7 | The combination of PMB with sorafenib synergistically suppresses HCC in CDX and PDX models. a** Representative images of tumors and effects of the indicated treatments on tumor growth in the HepG2, HCCLM3 and Hepa1-6 CDX mouse models (*n* = 8 mice per group). **b** Effects of the indicated treatments on tumor weights in the indicated CDX models (*n* = 8 mice per group). **c** The tumor formation efficiency of HCC cells harvested from HepG2 and HCCLM3 CDX-derived HCC tumors was evaluated by in vivo limiting dilution assay. (*n* = 8 mice per group). **d** Strategy for establishing PDX models from HCC patients. **e** The expression of MYC and SCARB2 were examined by WB in adjacent and HCC tissues from HCC patient of PDX model. Data are representative images from three independent experiments. **f** Representative images of tumors in the indicated PDX models (*n* = 8

mice per group). **g** Effects of the indicated treatments on tumor growth in the indicated PDX model (*n* = 8 mice per group). **h** Effects of the indicated treatments on the tumor weights in the indicated PDX models (*n* = 8 mice per group). **i** Flow cytometry analysis for ALDH activity using the ALDEFLUOR kit in PDX model with indicated treatments. **j** Schematic diagram illustrates that SCARB2 drives hepatic carcinoma initiation by supporting cancer stem cell traits and enhancing MYC transcriptional activity. **e, i** *n* = 3 biological repeats. Statistical significance was calculated by (**a, b, g, h, i**) two tailed Student's *t* test; (**c**) one-sided extreme limiting dilution analysis. Data are presented as means ± S.E.M. Source data are provided as a Source Data file.

## Animal studies

NOD-*scid* IL2Rg^null (NSG) mice (5–6 weeks old, male) were purchased from Shanghai Nan Fang Model Biotechnology Co. Ltd. (Shanghai, China). BALB/c nude mice (5–6 weeks old, female) and C57BL/6 mice (5–6 weeks old, male) were purchased from Beijing Hua Fu Kang Bioscience Co., Ltd. (Beijing, China). *Cre^Alb* (B6.Cg-Speer6-ps1Tg(Alb-cre)21Mgn/J) mice (5–6 weeks old, 1 male and 2 females; The Jackson Laboratory, 003574) were obtained from Cyagen Biosciences Inc. (Suzhou, China). *Scarb2* knockout (*Scarb2*^loxP/loxP, *Scarb2*^F/F) (5–6 weeks old, 1 male and 2 females) mice were generated by Cyagen Biosciences Inc. (Suzhou, China). The gRNA for the mouse *Scarb2* gene, the donor vector containing loxP sites, and *Cas9* mRNA were coinjected into fertilized mouse eggs to generate offspring with targeted conditional knockout. Mice with hepatocyte-specific conditional *Scarb2* knockout were generated by crossing *Scarb2*^F/F mice with *Cre^Alb* mice. *Myc*^F/F (C57BL/6JSmoc-*Igs*^2em1(CAG-LSL-Myc)Smoc)(5–6 weeks old, 1 male and 2 females, NM-KI-00039) mice were obtained from the Shanghai Research Center for Model Organisms (Shanghai, China). The conditional overexpression sequence with the CAG promoter-loxp-STOP-loxp-Myc-polyA structure was inserted into the H11 locus to establish the *H11-LSL-Myc* mouse model.

All mice were maintained in the animal facility at the Institute of Materia Medica under specific pathogen-free (SPF) conditions. Mice were housed in groups of 4–6 in individually ventilated cages on a 12 h light/dark cycle (07:30–19:30 light, 19:30–07:30 dark) in a room with controlled temperature ($23 \pm 2\,°C$) and relative humidity (40–50%). For animal studies, mice were earmarked before grouping and were then randomly separated into groups by an independent person; however, no particular method of randomization was used. All animal studies were approved by the Animal Experimentation Ethics Committee of the Chinese Academy of Medical Sciences (Permit No. IMB-20190423D702), and all procedures were conducted in accordance with the guidelines of the Institutional Animal Care and Use Committees of the Chinese Academy of Medical Sciences. The animal study was conducted in accordance with the Animal Research: Reporting of In Vivo Experiments (ARRIVE) guidelines.

## Viral library production

Human CRISPR Metabolic Gene Knockout library was a gift from David Sabatini (Addgene #110066). The workflow of this forward genetic screen is illustrated in Fig. 1a. First, 293FT cells were seeded at 15 cm culture dish (Corning Inc., #430599) for library scale production. Cells were transfected the next day at 80–90% confluency. 2.7 μg of pMD2.G (Addgene, #12259), 5.4 μg of psPAX2 (Addgene, #12260), 10.7 μg of lentivirus target plasmid (Addgene #110066) were transfected using 500 μl of DMEM and 877 μl of prepare polyethylenimine (PEI, Polysciences Inc., #23966). 5 h after transfection the media was changed. Virus supernatant was harvested 48 h post-transfection, filtered with a 0.45-mm PVDF filter (Millipore), aliquoted, and stored at −80 °C freezer until use.

## CRISPR/Cas9 knockout library screen

Around 10 million HCCLM3 cells were infected with lentivirus of Metabolic Gene Knockout library which contains targets 2,981 human metabolic genes with 30378 guide RNAs at a low MOI (~0.3) to ensure effective barcoding of individual cells. Then, the transduced cells were selected with 4 μg/ml of puromycin for 3 days to generate a mutant cell pool. The initial pool of 10 million cells was harvested for genomic DNA extraction as baseline. The remaining cells were precisely transferred to an ultra-low attachment 6-well plate at a ratio of 50000 cells per well. After 15 days of culture, tumorspheres larger than 70 μm (diameter >70 μm) and smaller than 40 μm (diameter <40 μm) were harvested by cell strainers (40 μm and 70 μm), and genomic DNA was extracted by Allprep DNA/RNA Mini Kit (QIAGEN) following the

manufacturer's protocol. The sgRNA library readout was performed using a two-steps PCR strategy. Firstly, the first PCR was used to amplify genomic DNA by using specific primers to the letiCRISPR-v1 vector with NEBNext® High-Fidelity 2X PCR Master Mix. PCR primers for library amplification: F-TTTCTTGGGTAGTTTGCAGTTTTAAAAT-TATGTTT;    R-TTGTGGATGAATACTGCCATTTGTCTCAAGATCTAG; Then the products from the first PCR were performed for second PCR to add appropriate sequencing adapters. PCR primers for next-generation sequencing (NGS): P5: AATGATACGGCGACCACCGA-GATCTACACCAGTCGACACTCTTTCCCTACACGACGCTCTTCCGATC TGACTATCATATGCTTACCGTAACTTG; P7: CAAGCAGAAGACGGCA-TACGAGATACATAGGTGACTGGAGTTCAGACGTGTGCTCTTCCGAT CTACTCGGTGCCACTTTTTCAAGTTGA; Finally, the products from the second PCR were subjected to massive parallel amplicon sequencing to determine sgRNA abundance, which was performed on an Illumina TruSeq by Annoroad Technology (Beijing, China). The significantly enriched sgRNA in HCCLM3 mutant cells, small tumorspheres compared to large tumorspheres were identified using the MAGeCK algorithm.

## Isolation and culture of tumor organoids

The noncancerous segment of the liver cancer tissue was removed, and liver cancer cells were isolated through a combination of mechanical disruption and enzymatic digestion. In brief, patient-derived or mouse HCC tissue of the appropriate size was minced, washed and incubated at 37 °C with digestion solution containing 1–2 mg/ml collagenase (Sigma, C9407) on an orbital shaker at 37 °C for 2–5 h. The suspension was strained through a 100 μm nylon cell strainer to retain tissue fragments. The cell pellet was obtained after centrifugation at $400 \times g$ for 5 min. The pellet was washed in cold advanced DMEM/F12 and was then mixed with 10 mg/ml cold Cultrex growth factor-reduced basement membrane extract (BME, Type 2, Pathclear) and allowed to solidify on prewarmed 24-well suspension culture plates at 37 °C for 30 min. After the BME was solidified, the samples were cultured in classical human liver organoid culture medium (advanced DMEM/F12 supplemented with 1% penicillin/ streptomycin, 1% GlutaMAX, 10 mM HEPES, 1:50 B27 supplement (without vitamin A), 1:100 N2 supplement, 1.25 mM N-acetyl-L-cysteine, 10% (vol/vol) Rspo-1 conditioned medium, 30% (vol/vol) Wnt3a-conditioned medium, 10 mM nicotinamide, 10 nM (Leu15)-gastrin I, 50 ng/ml EGF, 100 ng/ml FGF, 25 ng/ml HGF, 10 μM forskolin, 5 μM A8301, 25 ng/ml Noggin and 10 μM Y27632). The plates were transferred to humidified 37 °C/5% $CO_2$ incubators containing either 2% $O_2$ or at ambient $O_2$. The medium was changed every 4 days, and organoids were passaged every 1–4 weeks.

## Tumor organoid proliferation assay

After passaging for 7 days, tumor organoids were collected and passed through a 100 μm cell strainer to eliminate large organoids. Subsequently, organoids were resuspended in organoid medium and incubated with *CTRL*^cas9 or *SCARB2*^cas9 virus particles overnight at 37 °C/5% $CO_2$ incubators. Then organoids with or without *SCARB2* deletion were seeded at a density of 50 organoids /well in 96-well plates. Organoids were cultured for 1, 2, 3, 4 or 5 days. Organoid viability was assayed using CCK-8 solution (Dojindo, CK04) according to the manufacturer's instructions and results were normalized to Day1.

## Tumor organoid invasion assay

After passaging for 7 days, tumor organoids were collected. Subsequently, organoids were resuspended in organoid medium and incubated with *CTRL*^cas9 or *SCARB2*^cas9 virus particles overnight at 37 °C/5% CO2 incubators. Then organoids with or without SCARB2 deletion were collected and resuspended with 3.5 mg/ml rat tail collagen I (Ph 7.0). Subsequently, organoids with or without *SCARB2* deletion were seeded at a density of 30 organoids /well into 96-well plate and

Organoids with multicellular strands were counted as having protrusive invasion on day 4 after plating.

## CUT&Tag

CUT&Tag assay was performed according to the manufacturer's protocol using Hyperactive Universal CUT&Tag Assay Kit for Illumina (Vazyme, Nanjing, China, #TD903). In brief, HepG2 cells and HepG2 $SCARB2^{Cas9}$ cells were harvested, counted and centrifuged for 5 min at $2000 \times g$ at room temperature, 50000 cells per sample. The cells were washed with 500 μL Wash Buffer supplement with 1× Protease inhibitor cocktail and then resuspended with 100 μL Wash Buffer. Concanavalin A Beads were prepared with Binding buffer and 10 μL of activated beads were added per sample, invert to mix and incubate at room temperature for 10 min. Placed the tube on the magnetic stand and then discard the supernatant, the primary antibody incubation was performed on a rotating platform overnight at 4 °C. Beads were washed in Dig-wash Buffer for 3 times and incubated with goat anti rabbit secondary antibody for 1 h at a dilution of 1:100. After incubation, the beads were washed for 3 times in Dig-wash Buffer. Cells were incubated with hyperactive pA/G-transposon (pA/G-Tnp) at RT for 1 h and washed for 3 times in Dig-300 buffer to remove unbound pA/G-Tnp. Next, cells were re-suspended in 50 μL Trueprep Tagment Buffer L (TTBL) buffer and incubated at 37 °C for 1 h. The DNA fragments were extracted by adding 5 μL Proteinase K, 100 μL buffer L/B and 20 μL DNA Extract Beads at 55 °C for 1 h. Tubes were placed on a magnet stand to clear, then the liquid was carefully withdrawn. Without disturbing the beads, beads were washed twice with Buffer WA and Buffer WB. After allowing to dry ~5 min, 20 μL ddH$_2$O was added. The tubes were vortexed, quickly spun and allowed to sit for 5 min. Then the tubes were placed on a magnet stand and the liquid was withdrawn to a fresh tube.

To amplify libraries, 15 μL DNA was mixed with 25 μL 2 × CUT&Tag amplification mix, 5 μLof a universal N5, and 5 μL uniquely barcoded N7 primer, using a different barcode for each sample. The sample was placed in a Thermocycler with a heated lid using the following cycling conditions: 72 °C for 3 min; 95 °C for 30 min; 14 cycles of 98 °C for 10 s and 60 °C for 5 s; final extension at 72 °C for 1 min and hold at 4 °C. Post-PCR clean-up was performed by adding 100 μL VAHTS DNA Clean Beads (Vazyme, #N411), and libraries were incubated with beads for 5 min at RT, washed twice gently in 80% ethanol, and eluted in 20 μL ddH$_2$O for sequencing.

## CUT & Tag-sequencing analysis

The size distribution of libraries was determined by Agilent 2100 bioanalyzer, and the libraries were sequenced on Illumina NovaSeq6000 platform in paired-end reads following the manufacturer's instructions. Paired-end reads were aligned with *Homo_sapiens.GRCh38* genome using Bowtie2 version 2.2.5. Mapped reads were visualized using the Integrative Genomics Viewer (IGV).

## Chromatin immunoprecipitation (ChIP)-qPCR

ChIP assays were performed as per the manufacturer's protocol using a SimpleChIP® Plus Sonication Chromatin IP Kit #56383 (Cell Signaling Technology, Danvers, MA, USA). Briefly, cells were fixated with 1% formaldehyde for 10 min, incubated with glycine (50 mM final) 10 min and washed three times with PBS. After cell lysis and chromatin extraction, chromatin was sonicated to 100–500 bp using a Q800R3 Sonicator Chromatin and DNA Shearing system (Qsonica, USA), followed by centrifugation at 16,000 g for 10 min at 4 °C. Then the lysates were incubated overnight at 4 °C with ChIP grade antibodies specific to MYC (CST, #9402), which were then coupled to magnetic beads. Precipitated material was eluted (input Chromatin was used as control), the crosslink was reverted and DNA was purified by chloroform/phenol extraction and resuspended in DNA elution buffer. The eluted DNA was analyzed by qPCR. Species matched IgG (Cell Signaling Technology,

Danvers, MA, USA) was used as control for all ChIP experiments. The qPCR primer sequences were as follows: *hCAD* forward, 5′- ACGTG-GACCGACTCCGG-3′; *hCAD* reverse, 5′-CCATGGGAAGGGAACTCAGA-3′; *hCDK4* forward, 5′-AGGCATGTGTCATGTGTGATCTT-3′; *hCDK4* reverse, 5′- CCGCTCCCAGTCTTCCTTG-3′; *hHES1* forward, 5′-AATGA-GATCCGGAATCGGCG-3′; *hHES1* reverse, 5′-TCATCCGTAGGCTT-TAGGTTCTG-3′; *hLDHA* forward, 5′- ACGTCAGCATAGCTGTTCCA-3′; *hLDHA* reverse, 5′- AATGAGATCCGGAATCGGCG-3′; *hNCL* forward, 5′-TTGCGACGCGTACGAGCTGG-3′; *hNCL* reverse, 5′-ACTCCGAC-TAGGGCCGATAC-3′; *hPKM2* forward, 5′- GACTGATGGCGTAGCCT-3′; *hPKM2* reverse, 5′-ATAACCTTGAGGCTGA-3′; *Chr6* forward, 5′-TGGCATTGTCCTAATACTTCAGTGAT-3′; *Chr6* reverse, 5′- TTTCTGA AGTGCTGCTACCTCTCA-3′; *mCad* forward, 5′-CACTACGCTTAGGG CTCTGGCTTGC -3′; *mCad* reverse, 5′-GGGCCGCCATCGGGTCG-GAGCTGAG-3′; *mCdk4* forward, 5′- CCATGACACCGCCTTGTGCTC CACC -3′; *mCdk4* reverse, 5′-ATGGGAGGGGTGTGGTGGGGAAGGG-3′; *mHes1* forward, 5′- CGTTGTAGCCTCCGGTGCCCCGGGC -3′; *mHes1* reverse, 5′- CCAGCTCCAGATCCTGTGTGATCCG -3′; *mLdha* forward, 5′- TATTTACTGAAGGCCTGTTGCTTGC-3′; *mLdha* reverse, 5′- TCCACG TGTGCTGCGACACCCCAAA-3′; *mNcl* forward, 5′- TGTCATCACC-CAAGGCTGTGTGTGC-3′; *mNcl* reverse, 5′- GTTTGGTTTTCCCTTCAG-GAAAAAT -3′; *mPkm* forward, 5′-TAGGCCAGGGCAGATGGGGAGACCT-3′; *mPkm* reverse, 5′- GTGGCGGAAGGACATCGTAGACCAA -3′;

## Generation of the *MYC K148* mutation with the CRISPR/Cas9 system

The sgRNAs for introducing the *MYC K148R* knock-in mutation were annealed and ligated into the YKO vector. The gRNA sequences targeting *MYC K148R* were gRNA1: 5′-CAGCTTCTCTGAGACGAGCTTGG-3′ and gRNA2: 5′-TTTGCGCGCAGCCTGGTAGGAGG. The repair template designed with a homologous genomic flanking sequence was TCATCATCCAGGACTGTATGTGGAGCGGCTTCTCGGCCGCCGC-GAAGCTCGTCTCAGAGAGGCTGGCGTCCTACCAGGCTGCGCGCAAA-GACAGCGGCAGCCCGAACCCCGCCCGCGGCCA. 1 μg of each sgRNA plasmid was mixed with 1 μg of donor plasmid for transfection into HCCLM3 cells with Lipofectamine 3000 Reagent (Invitrogen, L3000008) according to the manufacturer's instructions. Twelve hours after transfection, HCCLM3 cells were treated with 4 μg/ml puromycin for 24 h. Then, the puromycin was removed from the cell culture medium, and the cultured cells were sorted into 96-well plates at 1 cell/well. The cells were incubated and expanded for 2–3 weeks, and all clones were further subjected to genomic DNA extraction (TIANamp Genomic DNA Kit, DP304), PCR amplification of the MYC sequence and Sanger sequencing. The sequences of the PCR primers were as follows: MYC-K148R forward, 5′-CTCGTCTCAGAGAGGCT GGCCTCCT-3′ and MYC-K148R reverse, 5′-AGGAGGCCAGCCTCTCTGA GACGAG-3′. The correct K148R knock-in cell clones were selected for further experiments.

## Cell culture

HepG2, Huh7, Hep3B, Hepa1-6, H22, 293FT, and HEK 293 T cells were purchased from Cell Resource Center, Peking Union Medical College. HCCLM3 cells were purchased from the China Center for Type Culture Collection (CCTCC). The cells had been authenticated by short tandem repeat (STR) profiling and characterized by mycoplasma detection and cell viability detection. HepG2 cells were cultured and maintained in MEM (Invitrogen, 11090081) supplemented with 10% fetal bovine serum (FBS; Invitrogen, CA, USA) and nonessential amino acids (Invitrogen, 11140050) under 5% carbon dioxide. HCCLM3, Huh7, Hepa1-6 and HEK 293 T cells were cultured and maintained in DMEM supplemented with 10% FBS. Hep3B cells were cultured and maintained in MEM-EBSS supplemented with 10% FBS. H22 cells were cultured and maintained in RPMI 1640 medium supplemented with 10% FBS. All cell lines were verified negative for mycoplasma contamination by MycoAlert™ Mycoplasma Detection Kit (Lonza, LT07-318).

## Generation of stable cell lines

To generate cells with stable *SCARB2* knockout, *SCARB2*[Cas9-1] and *SCARB2*[Cas9-2] lentiviral particles were purchased from TransOMIC Technologies Inc. and were then infected into HepG2 or HCCLM3 cells. The gRNA sequences targeting *SCARB2* were gRNA1: GTGTAGACCA-GAGTATCGAGAG and gRNA2: GGTGACCAGCGTCACGCTGCG. After 24 h of infection, stable cells were selected in medium containing 2 μg/ml or 4 μg/ml puromycin (Gibco, CA, USA) for 14 days. After 2–3 passages in the presence of puromycin, cultured cells were used for experiments without cloning.

## Generation of spontaneous HCC model mice

Hepatocarcinogenesis was induced by DEN as previously described. In brief, 14-day-old male *Cre^AlbScarb2*^F/F mice and *Cre^Alb* mice mice were injected intraperitoneally with DEN (25 mg/kg body weight). Mice were using cervical dislocation under anesthesia 8 months after DEN injection. *Cre^AlbMyc* mice with hepatocyte-specific *Myc* over-expression were generated by mating *Myc*^F/F mice with *Cre^Alb* mice. *Cre^AlbMyc* mice spontaneously developed liver cancer at approximately 2 months old.

To determine the role of *Scarb2* in hepatocarcinogenesis, *Cre^AlbScarb2*^F/F or *Cre^AlbScarb2*^F/+ mice with hepatocyte-specific *Scarb2* deletion were first generated by mating *Scarb2*^F/F mice with *Cre^Alb* mice. *Myc*^F/F mice were crossed with *Scarb2*^F/F mice to generate *Scarb2*^F/FMyc and *Scarb2*^F/+Myc mice, which were then crossed with *Cre^AlbScarb2*^F/+ mice to generate *Cre^AlbScarb2*^F/+Myc and *Cre^AlbScarb2*^F/FMyc trigenic mice. These mice were monitored for tumor incidence for 4 months.

## Establishment of PDX mouse models

Fresh HCC tissue was fragmented into small pieces (1–3 mm³) in medium. The tissue fragments were diluted with Matrigel (Corning, 354248) at a 1:1 ratio and mixed well by shaking. After NSG mice (5–6 weeks old, male) were shaved and anesthetized, the tissue mixture was implanted subcutaneously into these mice. Early passages (1–5) of primary tumor tissues from these PDX models were performed according to the above method.

## scRNA-seq assay

In scRNA-seq assay, human primary HCC cells isolated from tumor tissues of PDX mice were counted three times by three individuals independently and 15,270 live cells were finally used for 10× Genomics. The scRNA-Seq libraries were prepared with single cell 3' Library and Gel Bead Kit V3 (10x Genomics, 1000075), the cell suspension (300-600 living cells per microliter determined by Count Star) was loaded onto the Chromium single cell controller (10× Genomics) to generate single-cell gel beads in the emulsion according to the manufacturer's protocol. In short, single cells were suspended in PBS containing 0.04% BSA. Reverse transcription was performed on a S1000TM Touch Thermal Cycler (Bio Rad) at 53 °C for 45 min, followed by 85 °C for 5 min, and hold at 4 °C. The cDNA was generated and then amplified, and quality assessed using an Agilent 4200 (performed by CapitalBio Technology, Beijing). The libraries were finally sequenced using an Illumina Novaseq6000 sequencer with a sequencing depth of at least 100,000 reads per cell with pair-end 150 bp (PE150) reading strategy (performed by CapitalBio Technology, Beijing). The Cell Ranger Software Suite (Version 3.0.2) was used to perform sample de-multiplexing, barcode processing, and single-cell 3' UMI counting with *Homo_sapiens.GRCh38* as the reference genome. The filtered gene-barcode matrix was analyzed by PCA. Then Uniform Manifold Approximation and Projection (UMAP) was performed on the top 50 principal components for visualizing the cells. The MYC target gene score was calculated based on the expression of *CDK4, LDHA, CCNA1, CCND1, CCNE1, CCNE2, EIF4A, GLUT1*. The rescaled values for the MYC target genes were averaged and used as the MYC target gene score.

## Plasmid construction

Human *MYC*-Flag-tag (HG11346-CF), *SCARB2*-Flag-tag (HG11063-CF), *SCARB2*-Myc-tag (HG11063-CM), *MAX*-His-tag (HG12885-CH), *HDAC3*-HA-tag (HG11511-CY), and *SIRT1*-HA-tag (HG10830-NY) plasmids were purchased from Sino Biological Inc. (Beijing, China). The *MYC* truncations *MYC*-N (amino acids 1-144), *MYC*-C (amino acids 360–454), and the central region (amino acids 144-360) were inserted into the pEGFP-C1 vector by standard subcloning techniques. The *SCARB2* truncations *Scarb2*-N (amino acids 1-433), and *Scarb2*-luminal (amino acids 30-433) were inserted into the pcDNA3.1-myc-his vector (Invitrogen, V85520) by standard subcloning techniques.

## Luciferase reporter assay for MYC transcriptional activity

Cells were seeded in 12-well plates and transfected with Myc luciferase reporter plasmid (Yeasen Inc., #11544ES03) using Lipofectamine 3000 Reagent (Invitrogen, L300008). pTK-Renilla was used as the internal control. Luciferase activity was measured 20–24 h after transfection using a Dual-luciferase Reporter Assay System (Promega, USA).

## Single tumor cell preparation and sphere-forming culture

Human primary cells were separated with a Tumor Cell Isolation Kit, Human (Miltenyi Biotec, #130-095-929). The enzyme mix (2.2 ml of DMEM, 100 μl of Enzyme H, 50 μl of Enzyme R, 12.5 μl of Enzyme A) and 0.05–0.2 g of tumor samples with the fat, fibrous areas and necrotic areas removed were prepared. The tumor was cut into small pieces of 2–4 mm. The tissue pieces were transferred into a gentle MACS C tube containing the enzyme mix. Dissociation was initiated by running the gentle MACS program 37C_h_TDK_3. The sample was resuspended, and the cell suspension was filtered through a MACS Smart Strainer (40 μm or 70 μm) placed on a 50 ml tube. The cells and MACS Smart Strainer (40 μm or 70 μm) were washed with 20 ml of DMEM.

Cells were suspended in serum-free advanced DMEM/F12 supplemented with 100 IU/ml penicillin, 100 μg/ml streptomycin, 20 ng/ml human recombinant epidermal growth factor (hrEGF), 20 ng/ml human recombinant basic fibroblast growth factor (hrbFGF), 1% nonessential amino acids, 1% GlutaMAX, 2% B27 supplement (Invitrogen, USA), and 1% N2 supplement (Invitrogen, Carlsbad, CA, USA). Cells were subsequently cultured in ultra-low attachment 6-well plates (Corning Inc., Corning, NY, USA) at a density of no more than 10,000 cells/well.

## Passaging sphere culture

The sphere formation medium that contains non-sphere cells and sphere cells was collected in an Eppendorf tube and let this stand for 10 min at room temperature. Then the tumorspheres were washed with PBS and digested with Tryple Express Enzyme (Invitrogen, Carlsbad, CA, USA). When the majority of spheres were loose, PBS were added and gently triturated, centrifuged and the cells collect at 300 x g for 5 min. 2000 Cells/well were seeded in ultra-low attachment 6-well plates with conditioned medium.

## Limiting dilution assay in vitro

HepG2, HCCLM3, and Human primary HCC cells with or without *SCARB2* depletion, HepG2, and Human primary HCC cells with or without PMB treatment were plated in ultra-low attachment plates as the indicated dilution number (HepG2 and HCCLM3: 96, 48, 24, 12, 6 and 3 cells/well, human primary HCC cells: 250, 500, 1000, 2000, 5000, and 10000 cells/well; replicates of at least eight per dilution). The sphere-forming were observed using an inverted microscope. Extreme limiting dilution analysis was used to analyze data.

## Limiting dilution assay in vivo

Single-cell suspensions of primary HCC cells from the spontaneous HCC model in *Cre^AlbMyc*, *Cre^AlbScarb2*^F/+Myc or *Cre^AlbScarb2*^F/FMyc mice were sorted to obtain HCC cells, which were then injected into NSG

mice (5–6 weeks old, male). Single-cell suspensions of HCC cells from subcutaneous HCC tumors were isolated to obtain HCC cells, which were then injected into BALB/c nude mice (5–6 weeks old, female). The outgrowths were analyzed at 8 weeks post transplantation. The tumorigenicity of the transplanted cell suspension was calculated using an extreme limiting dilution assay (ELDA).

## Mouse models of tumor growth and metastasis

To evaluate the effect of combined sorafenib treatment and *SCARB2* knockout on the inhibition of tumor growth and metastasis, BALB/c nude mice (5 to 6 weeks old, male) were earmarked before grouping and were then randomly separated into four groups ($n = 8$ per group) in a blinded manner by an independent person. The tumor growth model in BALB/c-nude mice was established by subcutaneous (s.c.) injection of $2 \times 10^6$ HepG2 cells or HepG2 *SCARB2*^cas9 cells diluted with 100 µl of Matrigel (Corning, 354230) at a 1:1 ratio. Tumor growth was monitored, and the tumor size was measured using calipers. To establish the mouse model of experimental metastasis, BALB/c-nude mice were injected via the tail vein with $2 \times 10^6$ HCCLM3 cells or HCCLM3 *SCARB2*^cas9 cells. To evaluate the survival rate, these mice were monitored for more than 3 months. The number of lung metastases was determined.

## Immunoprecipitation, immunoblotting, immunostaining and HCC tissue microarray analysis

Co-IP was performed as described previously. Cells were harvested and lysed for 30 min in Co-IP buffer supplemented with a complete protease inhibitor (Cell Signaling Technology, 5817), 1 mM trichostatin A (Selleck, S1045), and 5 mM nicotinamide (MedChemExpress, HYB0150) on ice. Centrifugation was performed to obtain the supernatant, which was then incubated first with the indicated antibodies at 4 °C overnight, and then with Protein A/G Plus-Agarose (Santa Cruz Biotechnology, TX, USA) at 4 °C for 2 h. Soluble lysates were incubated with the indicated anti-Myc magnetic beads (Bimake.com, B26302), anti-Flag affinity gel (Bimake.com, B23102) or anti-HA affinity gel (Bimake.com, B23302) at 4 °C overnight. Complexes were eluted from the beads and were then boiled for 10 min. The precipitated proteins were subjected to SDS-PAGE and immunoblotting with the corresponding antibodies. For immunoblot analysis, proteins were extracted from cells and liver tissues using RIPA buffer (Cell Signaling Technology, MA, USA). The protein concentrations were determined with a BCA Protein Assay Kit. Protein extracts were separated by SDS-PAGE, transferred onto PVDF membranes, and subjected to immunoblot analysis. Images of the Western blots were acquired with a Tanon 5200 chemiluminescent imaging system (Tanon, Shanghai, China). Western blots images were analyzed by Gel Pro Analyzer 3.2.

For immunofluorescence and colocalization assays, cells were seeded in 12-well plates and processed differently according to the experimental requirements. Next, the cells were briefly washed with PBS and fixed with 3.7% formaldehyde in PBS for 10 min, washed three times with PBS and permeabilized with 0.5% Triton X-100 for 15 min. Then, the cells were washed with PBS three times and blocked with 3% bovine serum albumin (BSA) for 1 h at 37 °C. Samples were incubated with primary antibodies overnight at 4 °C and with secondary antibodies for 2 h. Nuclei were stained with 4′,6-diamidino-2-phenylindole (DAPI) in blocking buffer. Images were acquired using a confocal microscope (Olympus Microsystems, Fv3000, CA, USA). Quantitative image analysis was performed with Imaris 9.3.1 software. The Pearson correlation coefficient was used to analyze colocalization between two target proteins.

For immunohistochemical analysis, tissue sections were deparaffinized in xylene and rehydrated through a graded alcohol series and distilled water. Antigen retrieval was carried out in a microwave with citrate buffer (10 mM sodium citrate buffer, pH 6.0) at a subboiling temperature for 15 min. Sections were permeabilized with 0.5% Triton

X-100 in PBS for 20 min. Endogenous peroxidase activity was blocked with 3% $H_2O_2$ solution for 10 min, and the sections were then washed three times with PBS. Blocking buffer (3% BSA/PBS) was added to the sections and incubated for 30 min. The sections were then incubated with the indicated primary antibodies at 4 °C overnight. After washing three times, the sections were incubated for 30 min with the corresponding secondary antibodies at room temperature. Signals were detected with freshly made DAB substrate solution (ZSGB-BIO Company, Beijing, China). The sections were then counterstained with hematoxylin, dehydrated, and mounted with coverslips. Images were acquired using an Olympus DP72 microscope (Olympus Microsystems, CA, USA) and analyzed with Image-Pro Plus 5.1.

For immunoblotting, the following antibodies were used: anti-GAPDH (ZSGB-BIO TA-08, 1:2000), anti-MYC (D3N8F) (CST, #13987 S, 1:1000), anti-MYC (CST, #9402 S, 1:1000), anti-Max (S20) (CST, #4739, 1:1000), anti-acetylated lysine (CST, #9441, 1:1000), anti-HDAC3 (7G6C5) (CST, #3949, 1:1000), anti-SCARB2 (Abcam, ab176317, 1:1000), anti-GCN5 (Abcam, ab217876, 1:1000), anti-BRD4 (CST, #13440, 1:1000), anti-KAT5 (Abcam, ab151432, 1:1000), anti-Cyclin A2 (E1D9T) (CST, #91500 S, 1:1000), anti-Cyclin D (CST, #2922, 1:1000), anti-Cyclin E1 (D7T3U) (CST, #20808, 1:1000), anti-p300 (D8Z4E) (CST, #86377, 1:1000), anti-CDK4 (D9G3E) (CST, #12790, 1:1000), anti-N-Cadherin (D4R1H) (CST, #13116, 1:1000), anti-EIF3A (D51F4) (CST, #3411, 1:1000), anti-E2F2 (Abcam, ab138515, 1:1000), anti-IRP2 (D6E6W) (CST, #37135, 1:1000), anti-MMP14 (Abcam, ab51074, 1:1000), anti-Lamin A/C (4C11) (CST, #4777, 1:1000), anti-Na, K-ATPase (D4Y7E) (CST, #99935, 1:1000), anti-Myc-tag (MBL, #562, 1:1000), anti-GFP-tag (MBL, #598, 1:1000), anti-DDDDK-tag (MBL, PM020, 1:1000), anti-HA-tag (MBL, #561, 1:1000), and anti-His-tag (MBL, PM032, 1:1000). For immunofluorescence and immunohistochemistry, the following antibodies were used: anti-MYC (R&D, AF3696, 1:100), anti-MYC (R&D, MAB3696, 1:100), anti-MYC (Novus, NB600-302, 1:100), anti-SCARB2 (Abcam, ab176317, 1:100), anti-HDAC3 (Novus, NB500-126, 1:100), Alexa Fluor 488 (Thermo Fisher, R37114, 1:200), Alexa Fluor 488 (Thermo Fisher, R37118, 1:200), Alexa Fluor 488 (Abcam, ab150173), Alexa Fluor 555 (Thermo Fisher, A-31572,1:200), Alexa Fluor 555 (Thermo Fisher, A-31570, 1:200), Alexa Fluor 555 (Thermo Fisher, A-21432, 1:200), Alexa Fluor 647 (Thermo Fisher, A-31571, 1:200), and Alexa Fluor 647 (Thermo Fisher, A-31573, 1:200). For Cut & Tag, the following antibody was used: c-Myc Antibody (CST, #9402 S, 3 µg/sample).

## Flow cytometry and cell sorting

For Flow cytometry, cell suspensions from HCC cell lines, primary human HCC cells or primary mouse HCC cells were directly labeled. Fluorescently labeled antibodies against the following surface proteins were used for cell staining: anti-LIMPII (ab176317, Abcam, 1:100), followed by staining with secondary antibodies labeled with FITC. PE anti-human CD13 (Cat. 301704, BioLegend, 1:100), APC/Cyanine7 anti-human CD24 (Cat.311132, Biolegend, 1:100), APC anti-human CD133 (Cat. 394009, BioLegend, 1:100), FITC anti-human EpCAM (Cat. 369813, BioLegend, 1:100), PE/Cyanine7 anti-human EpCAM (Cat. 324222, BioLegend, 1:100), PE anti-mouse CD24 (Cat. 101807, Biolegend, 1:100), PE/Cyanine7 anti-mouse CD133 (Cat. 141209, BioLegend, 1:100), and APC anti-mouse EpCAM (Cat. 118214, BioLegend, 1:100). Then, cells were washed, and data were acquired using a BD FACS Verse or CytoFLEX flow cytometer and analyzed with FCS EXPRESS or FlowJo 10.8.1 software. For cell sorting, cell suspensions from HCC cell lines and primary human HCC cells were stained with PE anti-human CD13 and APC anti-human CD133, followed by being sorted CD133⁺CD13⁺ and CD133⁻CD13⁻ cells with FACS Aria III (BD Bioscience). After sorting, CD133⁺CD13⁺ and CD133⁻CD13⁻ cells infected with *CTRL*^Cas9 or *SCARB2*^Cas9 virus particles were seeded in ultra-low attachment 96-well plates or 6-well plates with sphere-forming culture medium at a density of 3000 cells/well. Then the proliferation of

CD133$^+$CD13$^+$ cells and CD133$^-$CD13$^-$ cells with or without *SCARB2* knockout was evaluated by Cell Counting Kit-8 (CCK-8) assay. The expression of stem markers (*CD24* and *EpCAM*) and stem transcription factors (*NANOG, SOX2, and OCT4*) in CD133$^+$CD13$^+$ cells with or without *SCARB2* knockout were detected by qPCR.

## Cell Counting Kit-8 (CCK-8) assay

For the CCK-8 assay, HepG2 and HCCLM3 cells with or without *SCARB2* deletion were seeded at a density of 1000 cells/well in 96-well plates. Cells were cultured for 1, 2, 3, 4 or 5 days. To generate the time effect curve of sorafenib treatment, *CTRL$^{cas9}$* and *SCARB2$^{cas9}$* cells were seeded in 96-well plates at a density of 2000 cells/well. Vehicle (DMSO, Merck, D2650) or sorafenib (15 μM)-treated cells were cultured for 1, 2, 3, 4 or 5 days. To generate the dose effect curve of sorafenib treatment, vehicle (DMSO)- or sorafenib (2, 4, 8, 16, 32, 62.5, 125, 250, 500 μM)-treated cells were cultured for 24 h. To evaluate the antiproliferative effects of combined treatment with PMB and sorafenib, HepG2, H22, Hepa 1–6, Huh7, Hep3B and HCCLM3 cells were seeded in 96-well plates at a density of 10000 cells/well. Cells were treated with sorafenib (15 μM) and an appropriate concentration of inhibitor and cultured for 24 h. Subsequently, 10 μl of CCK-8 solution (Dojindo, CK04) was added to each well, and the plates were incubated at 37 °C for 2 h. Finally, the absorbance was measured at 450 nm.

## Invasion assay

Transwell assays were performed using Millicell inserts (8.0 μm, Millipore, Billerica, MA, USA) to evaluate cell invasion. Millicell inserts were precoated with 10 μg/ml fibronectin and Matrigel (1:8; BD Bioscience, Bedford, MA, USA) and then allowed to dry. Millicell inserts were placed into the wells of a 24-well plate containing culture medium supplemented with 10% FBS. Cells ($5 \times 10^4$ cells/well) were starved overnight and were then seeded in the upper chambers without FBS culture medium. In total, 12 h later, the migrated cells were fixed with paraformaldehyde, stained with 0.1% crystal violet and counted.

## Real-time PCR and RNA interference

Total RNA was extracted using TRIzol (Invitrogen, CA, USA) according to the manufacturer's instructions. Total cellular RNA was reverse transcribed using oligo(dT) primers and M-MLV reverse transcriptase (Transgen Biotech, Beijing, China). PCR was performed using a MyCycler thermal cycler and analyzed using qTOWER. The PCR primer sequences were as follows: *Gapdh* forward, 5′-CATCACTGCCACCCA-GAAGACTG-3′; *Gapdh* reverse, 5′-ATGCCAGTGAGCTTCCCGTTCAG-3′; *Tfap4* forward, 5′-GACGCGAGATTGCCAACAGCAA-3′; *Tfap4* reverse, 5′-TGCTGTCTGCTGGAGAATGGCT-3′; *Pld6* forward, 5′-TCTGCCTCTTCG CCTTCTCCAG-3′; *Pld6* reverse, 5′-GTAGTCGCAGTCAGTGATGACC-3′; *Glut4* forward, 5′-GGTGTGGTCAATACGGTCTTCAC-3′; *Glut4* reverse, 5′-AGCAGAGCCACGGTCATCAAGA-3′; *Glut1* forward, 5′-GCTTCTCCA ACTGGACCTCAAAC-3′; *Glut1* reverse, 5′-ACGAGGAGCACCGTGAA-GATGA-3′; *Cdk4* forward, 5′-CATACCTGGACAAAGCACCTCC-3′; *Cdk4* reverse, 5′-GAATGTTCTCTGGCTTCAGGTCC-3′; *Ccne1* forward, 5′-AAGCCCTCTGACCATTGTGTCC-3′; *Ccne1* reverse, 5′-CTAAGCAGC-CAACATCCAGGAC-3′; *Ccnd1* forward, 5′-GCAGAAGGAGATTGTG CCATCC-3′; *Ccnd1* reverse, 5′-AGGAAGCGGTCCAGGTAGTTCA-3′; *Ccna1* forward, 5′-GCTACTGAGGATGGAGCATCTG-3′; and *Ccna1* reverse, 5′-CAGCTTCCAGAAGGCTCAGTTC-3′. *CD24* forward, 5′-CACGCAGATTTATTCCAGTGAAAC-3′;*CD24* reverse, 5′-GACCACGAA-GAGACTGGCTGTT-3′; *EpCAM* forward, 5′-GCCAGTGTACTTCAGT TGGTGC-3′; *EpCAM* reverse, 5′-CCCTTCAGGTTTTGCTCTTCTCC-3′; *SOX2* forward, 5′-GCTACAGCATGATGCAGGACCA-3′; *SOX2* reverse, 5′-TCTGCGAGCTGGTCATGGAGTT-3′; *NANOG* forward, 5′- CTCCAA-CATCCTGAACCTCAGC -3′; *NANOG* reverse, 5′- CGTCACACCATTGC TATTCTTCG -3′; *OCT4* forward, 5′-CCTGAAGCAGAAGAGGATCACC-3′; *OCT4* reverse, 5′- AAAGCGGCAGATGGTCGTTTGG-3′; *SCARB2* forward, 5′-GCCAATACGTCAGACAATGCCG-3′; *SCARB2* reverse, 5′-CTCAT

CTGCTTGGTAAAAGTGTGG -3′; *GAPDH* forward, 5′- GTCTCCTCT GACTTCAACAGCG -3′; *GAPDH* reverse, 5′-ACCACCCTGTTGCTG-TAGCCAA-3′; TIP60 siRNA (sc-37967), GCN5 siRNA (sc-37947), and p300 siRNA (sc-29432) were purchased from Santa Cruz Biotechnology.

## Structured illumination microscopy (SIM)

Cells (~20,000) were grown on coverslips and fixed with formalin after the indicated treatment. Subsequently, the cells were washed in Tris-buffered saline (TBS) and permeabilized for 5 min with 0.5% Triton X-100 in TBS. The reaction was quenched for 10 min with 50 nM glycine in TBS, and blocking was performed for 30 min with 5% normal mouse serum or normal goat serum in a 0.2% gelatin-TBS solution depending on the isotype of the secondary antibody. Subsequently, 30-μl droplets containing the indicated dilutions of antibodies specific for MYC (R&D, AF3696, 1:100), SCARB2 (Abcam, ab176317, 1:100) or HDAC3 (Novus, NB500-126, 1:100) were placed on Parafilm in a dark humidified chamber. Coverslips were placed face down on the droplets and incubated overnight at 4 °C. The next day, the coverslips were lifted by adding a small volume (200 μl) of TBS under the coverslip. After washing 3 × 20 min with 1 ml of 0.2% gelatin-TBS, the coverslips were incubated with Alexa Fluor 488 (Thermo Fisher, R37114, 1:200) and Alexa Fluor 555 (Thermo Fisher, A-31572, 1:200) secondary antibodies as described for primary antibody incubation for 1 h at room temperature. After washing 3 × 10 min with 1 ml of 0.2% TBS-gelatin and 1 wash with regular TBS, the coverslips were mounted using soft-set mounting medium with DAPI (Vectashield) and sealed with nail polish. Images were acquired using a confocal microscope (Olympus Microsystems, CA, USA) or a GE Healthcare 3D structured illumination microscope. Intensity plots of individual pixels taken from a straight line in the indicated immunofluorescence images were generated by the twin slicer tool in Imaris 3D & 4D imaging software (Bitplane AG, Switzerland). Images were cropped and processed in Adobe Photoshop. When comparisons were made between images from the same experiment, all levels were adjusted equally, and the ratio between the levels was not altered.

## Mouse models to evaluate tumor growth and combination treatment with sorafenib and PMB

To evaluate the effect of combination treatment with sorafenib and PMB (polymyxin B sulfate) on tumor growth, BALB/c nude (5–6 weeks old, female), NSG mice (5–6 weeks old, male) or C57BL/6 (5–6 weeks old, male) were earmarked before grouping and were then randomly separated into four groups (*n* = 8 per group) in a blinded manner by an independent person. To establish the tumor growth model in BALB/c nude mice, $2 \times 10^6$ HepG2 or HCCLM3 cells per mouse diluted with 100 μl of Matrigel (Corning, 354230) at a 1:1 ratio was subcutaneously injected into the right flanks of nude mice. To establish the PDX model, tumor masses (0.1 cm$^3$) were subcutaneously transplanted into NSG mice. To establish the tumor growth model in C57BL/6 mice, $6 \times 10^6$ Hepa 1–6 cells per mouse diluted with 100 μl of Matrigel (Corning, 354230) at a 1:1 ratio was subcutaneously injected into the right flanks of the mice. To establish the HCC tumor spheroids model in BALB/c nude mice, the sphere formation is used to enrich CSCs from HCCLM3 cells. Tumor spheroids were collected and digested to single cell. Then the $1 \times 10^5$ HCC CSCs infected with *CTRL$^{cas9}$* or *SCARB2 $^{Cas9}$* virus particles per mouse diluted with 100 μl of Matrigel (Corning, 354230) at a 1:1 ratio was subcutaneously injected into the right flanks of the mice. Tumors were allowed to grow for 7 days, and mice were then administered vehicle (Kolliphor® HS 15, *i.g.*, once a day), sorafenib (30 mg/kg, *i.g.*, once a day), PMB (25 mg/kg, *i.g.*, once a day), or PMB plus sorafenib for 14 days. During the indicated treatments, tumor burden was evaluated by measuring tumor volumes to determine the therapeutic efficacy in the mouse models and PDX models. Tumor volume was monitored thrice a week and calculated as follows: tumor volume

= 0.5 x L x W$^2$, with L and W, as the largest and smallest diameters, respectively. All animal studies were approved by the Animal Experimentation Ethics Committee of the Chinese Academy of Medical Sciences. The tumor size didn't exceed 20 mm in any direction, which was permitted by our institutional review board. Mice were immediately euthanized when tumor size exceeded 20 mm by the last day of measurement in any direction. At the endpoint, animals were euthanized using cervical dislocation under anesthesia as approved by CULTAR.

### In vitro coimmunoprecipitation

Purified His-tagged MYC (TP760019) was purchased from Origene Inc. Purified His-tagged SCARB2 (11063-H03H) and Purified GST-tagged HDAC3 (11511-H20B) were purchased from Sinobiological Inc. Purified His-tagged MYC (5 μg) and His-tagged SCARB2 (5 μg); Purified His-tagged MYC (5 μg) and GST-tagged HDAC3 (5 μg) were mixed in a reaction buffer (100 μl) consisting of 1% NP-40, 120 mM NaCl, 40 mM tris-HCl (pH 7.4), 1.5 mM sodium orthovanadate, 50 mM sodium fluoride, 10 mM sodium pyrophosphate, and protease inhibitor cocktail (Roche) for 6 h at 4 °C. Co-IP experiments were performed as described in the section of immunoprecipitation.

### Proximity ligation assay (PLA)

A total of $1 \times 10^4$ cells was seeded overnight in a 24-well plate. The next day, cells were washed, fixed in 4% paraformaldehyde solution, permeabilized with 0.5% Triton-X100, blocked with Duolink® Blocking Solution, and probed with antibodies directed against MYC (R&D, NB600-302, 1:100) and MAX (Cell Signaling Technology, #4739, 1:100). The cells were then treated with the Duolink In Situ Red Starter Mouse/Rabbit kit (Sigma Aldrich, DUO92101-1KT) according to the manufacturer's instructions. Images were captured with an OLYMPUS confocal microscope.

### GSEA

We ranked genes by their association with $Cre^{Alb}Scarb2^{F/F}Myc$ mice vs. $Cre^{Alb}Myc$ mice and HepG2 cells vs. PMB treated HepG2 cells using the signal-to-noise metric determined by GSEA according to the log2 fold change values. MYC target gene sets were obtained from a database (http://software.broadinstitute.org/gsea/msigdb/index.jsp).

### Surface plasmon resonance analysis

A BIAcore T200 system (GE Healthcare) was used to analyze the surface plasmon resonance binding kinetics between SCARB2 and the indicated small molecules. In brief, SCARB2 protein (Sino Biological, China) was immobilized onto channel 2 in a CM5 sensor chip (GE Healthcare) through a standard coupling protocol. To measure the binding kinetics, the indicated small molecule in twofold serial dilutions and a buffer blank for baseline subtraction were sequentially injected, with a regeneration step (glycine, pH 2.5) performed between each cycle. The equilibrium dissociation constant was calculated with Bia Evaluation Software 4.1.

### Quantitative proteomics

Whole-cell lysates of HCCLM3 cells were immunoprecipitated with an anti-MYC antibody (CST, 9402 S), anti-SCARB2 antibody (Abcam, ab176317) or IgG1 isotype control antibody (CST, 5415) and Protein A/G Plus-Agarose (Santa Cruz Biotechnology) overnight at 4 °C. Interaction complexes were eluted from the beads by heating at 98 °C for 10 min. MS-based Quantitative proteomics and analysis were performed by Beijing Qinglian Biotech Co., Ltd. For analysis, MS raw data were processed with Proteome Discoverer 2.4. Trypsin/P was specified as the cleavage enzyme allowing up to 2 missing cleavages. The searched parameters were set as follows: mass tolerance for precursor ion was 15 ppm and mass tolerance for production was 0.02 Da. Carbamidomethyl residues on Cys were specified as the fixed modification and the oxidation on Met, acetylation of the N-terminus. Search results were filtered with 1% FDR at both protein and peptide levels.

### Statistics & reproducibility

No data were excluded from the analyses. For in vivo cancer studies, mice were randomly assigned into experimental groups. The Investigators were not blinded to allocation during experiments and outcome assessment. Comparisons between two groups were performed by unpaired Student's $t$ test. Pearson correlation analysis was used to determine correlations between groups. The Kaplan–Meier method was used to analyze survival. All data are expressed as the mean ± standard error of the mean (S.E.M) values. Generally, all experiments were carried out with $n \geq 3$ biological replicates. $P < 0.05$ was considered statistically significant. All the data presented has been reviewed by a statistician to ensure scientific rigor and diligence of our data. Analyses were performed using GraphPad Prism 9.0 software.

### Reporting summary

Further information on research design is available in the Nature Portfolio Reporting Summary linked to this article.

## Data availability

The RNA-seq, scRNA-seq and CUT&Tag data generated in this study have been deposited in the NCBI Gene Expression Omnibus (GEO) database under accession code GSE185844 and GSE207673. The mass spectrometry proteomics data generated in this study have been deposited to the ProteomeXchange Consortium (http://proteomecentral.proteomexchange.org) via the iProX partner repository under accession code PXD044544. The datasets from cancer genome atlas (TCGA) were analyzed using the UALCAN platform[28] (https://ualcan.path.uab.edu/analysis.html). *SCARB2* mRNA expression in sorafenib-resistant HCC cell and their parental cells was analyzed on the following accession code GSE121153. Pearson's correlation between *SCARB2*, *SCARB1* or *CD36* expression with the known CSC markers gene was analyzed on Gene Expression Profiling Interactive Analysis (GEPIA) database[29] (http://gepia.cancer-pku.cn). The uncropped blot figures, and original data underlying Figs. 1–7 and Supplementary Figs. 1–6 are provided as a Source Data file. The remaining data are available within the Article, Supplementary Information or Source Data file. Source data are provided with this paper.

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

## Acknowledgements

This work was supported by grants from the National Key R&D Program of China (2022YFA1106100), the National Natural Science Foundation of China (82222070, 82273973, 82073887, 81872904 to K.L., 82003798 to F.W.), the "Ten thousand plan"—National high-level talents special support plan to K.L., the CAMS Innovation Fund for Medical Sciences (2021-I2M-1-030 to K.L.; 2021-I2M-1-070 to T.-t.Z.), the CAMS Innovation Engineering Platform Fund for Medical Sciences (2022-I2M-2-002 to

K.L), and the Fundamental Research Funds for the Central Universities (2022-RC350-07, 3332022149).

## Author contributions

K.L., J.D.J. and J.Y. conceptualized the study and participated in the overall design, supervision and coordination of the study. F.W., Y.G. and S.T.X. designed and performed most of the experiments. L.Y.Z., H.M.J., T.T.Z., Y.X.L., C.X.Z. F.W., T.N.S.Q., and Y.L. participated in the molecular and cellular biological experiments. J.W. and Y.C.Y. performed the animal studies. K.L., J.D.J. and J.Y. wrote the manuscript. All authors read and approved the manuscript.

## Competing interests

The authors declare no competing interests.
