## [Peer Review File · Nature Communications]

Reviewers' Comments:

Reviewer #1:

Remarks to the Author:

In this manuscript, Wang et al., report a role for the lysosomal membrane protein SCARB2 in driving cancer stem cells in hepatocellular carcinoma (HCC). They conduct a focused CRISPR knockout screen in human HCC tumorspheres and identify a number of genes required for their maintenance, one of which is SCARB2. They show that SCARB2 is overexpressed in HCC, and that its overexpression correlates with poor patient survival. They also show that Scarb2 deletion suppresses tumor initiation and progression in genetically engineered mouse models of liver cancer. By transcriptional profiling, they show that canonical Myc target gene sets are suppressed in the Scarb2 null setting, and present evidence that loss of Scarb2 is associated with decreased binding of Myc to chromatin, as well as decreased acetylation of Myc on lysine 148 (K148). They further present evidence that SCARB2 promotes acetylation of MYC at K148 by antagonizing the effects of HDAC3. By *in silico* screening, they identify the FDA-approved drug polymyxin B (PMB; usually used to treat conjunctivitis) as an inhibitor of the SCARB2–MYC interaction, and show that it has anti-tumor activity *in vivo* in combination with sorafenib.

This is a very interesting and potentially very important study. The identification of SCARB2 as a driver of cancer stem cells in HCC is an important finding, as are the mechanistic connections to MYC, and identification of PMB as an agent with significant anti-tumor activity *in vivo*. It is easy to see how these findings could lead to a new way to target MYC (a prized but undruggable drug target) in liver cancer. The work itself is quite exhaustive and comprehensive, of generally very high quality, and interpreted fairly. Key conclusions are backed up by multiple layers of evidence, and it is difficult to find fault with most of the approaches or the conclusions drawn from the totality of the work that is presented.

That said, one area that should be stronger, and needs to be strengthened, are the mechanistic connections between SCARB2, HDAC3, and MYC.

In Figure 3, the authors show that Scarb2 deletion suppresses Myc target genes in mice. But the accompanying western blot (Fig. S3b) shows that Myc levels are lower in these mice. Rather than being that Scarb2 controls the binding of Myc to chromatin, as they later show from cell based assays, this could mean that *in vivo* Scarb2 controls Myc levels. This discrepancy raises the concern that Scarb2 is acting differently *in vitro* and *in vivo* and as Myc levels could simply change as a result of the decreased proliferation of the cancer cells, this raises concern that the mechanistic model built from cell line studies is incorrect.

Figure 3 also shows that SCARB2 deletion decreases the binding of MYC to chromatin, and this then becomes the proposed mechanism of action. They go on to claim that this is due to control of acetylation of MYC at K148 (by interfering with HDAC3 interaction with MYC). But they never show that the K148R mutant of MYC has altered chromatin binding properties (hard to imagine given how far away this residue is from the DNA binding domain of MYC); they do not show that SCARB2 and HDAC3 directly affect MYC binding to chromatin; they do not show that PMB alters MYC binding to chromatin. Indeed, they argue that SCARB2 influences the interaction of MYC with chromatin co-factors—not proteins that impact MYC binding)—creating a confusing hole in their argued mechanism of action. Further, there is no evidence that SCARB2 or HDAC3 interacts directly with MYC, which creates further weaknesses in the model.

Reviewer #2:

Remarks to the Author:

In this manuscript by Wang et al., the authors aim at identifying drivers of HCC cancer stem cell development and potential therapeutics that eliminate HCC. For that, the authors perform a metabolic CRISPR/Cas9 screen and identify Scarb2 KO reduces HCC development. They suggest that Scarb2 maintains CSC state. To prove that mechanistically, they go one step further and describe that the mechanism relates to SCARB2 binding to MYC which inhibits MYC deacetylation in a HDCA3-depenedent manner and subsequently triggers MYC activation.

The manuscript is a great effort, a tour de force to try to investigate the role of this membrane lysosomal protein in HCC. The authors use all types of in vitro models, from spheroids to tumour organoids and cell lines as well as in vivo mouse models. While there is a lot of work, my main concerns lie on the conclusion that SCARB2 is important for cancer stem cells, which is mainly drawn from the use of cancer cell lines that have been long term expanded in culture., or from a mouse model that deletes the gene in development. The human data is nice but correlative. In addition, the interaction with MYC seems interesting but poses questions. My main comments are below.

Major comments

1) SCARB2 as a critical gene for maintaining the stemness of HCC cells

the main conclusion that SCARB2 plays a role in HCC CSC viability comes from the screen in cell lines. While this reviewer agrees that this protein plays a role in these cell lines, not proven that this is through an effect on CSC. There are some conclusions here that do not seem to match between the cell lines and the primary tissue, or tumor organoids used in figure 1J. In fact cancer cell lines are homogenous and, contrary to organoids, would not contain CSC and differentiated cells. This conclusion could only be obtained if the authors can identify CSC in their cultures and then compare them to non-CSC and observe that SCARB2 plays different roles in the two populations. Since SCARB2 mutant organoids are nicely viable (according to the live-death staining) and only when treated with Sorafenib they die, that would rather suggest the opposite, that SCARB2 KO affects the non-proliferative non-CSC pool in organoids. One potential experiment to address that would be that the authors stain the WT and mutant untreated organoids for CSC markers and observe a reduction of these. This could also be done by FACS for more quantitative analysis. In addition proof that it is the CSC pool in the cell lines the one affected would be necessary to sustain that claim.

Similarly, the mouse data (depletion of SCARB2 in AlbCre mice) indicates that there is a reduction in tumour size in these mutants. However, the conclusion that this is driven by effect specific on the CSC pool is not proven. Mainly AlbCre deletes in the embryo, at then hepatoblast stage, and that is not a "cancer stem cell". In addition, the model would be artificial, as the deletion would occur in development and not in adulthood, as it is supposed to be for a somatic disease such as cancer. Also, note that the authors insist on mentioning "Suppress, tumour" e.g. in. "Scarb2 deletion suppresses the initiation and progression of HCC." However, the correct wording is reduce tumour growth and tumour metastasis, since the mutant cells and mice develop tumours but these are smaller, or when mutant cells are injected these still generate metastasis, although less (as seen in figure 2D-K), the conclusion that "suppress is . This should be amended through the the text.

Mainly there is a correlation between loss of this gene and overall decrease in tumour growth but there is no proof that this is linked to CSC reduction.

2) Many figures suffer from lack of information on number of biological and technical replicates used. Mainly, except for the number of human samples in Fig 1 and mouse experiments this is missing in all in vitro experiments.

In addition, the statistical analysis is generalized on a methods section in the manuscript. Unfortunately, this precludes the evaluation whether the statistical tests are correct. Mainly, the analyses of all figures where there are multiple biological replicates with multiple technical replicates should follow a t-test of means if data is normally distributed, which has not been addressed here. I suggest authors to follow the directions published by Lord et al., J. Cell Biol. 2020 Vol. 219 No. 6 or Pollard et al., Molecular Biology of the cell Volume 30 June 1, 2019 among others.

3) Myc and SCARB2 interaction.

The authors aim at investigating the mechanism by which myc tumours are reduced in the absence of SCARB2. They observe a clear correlation between the mutant cells and an increase in MYC targets. This is interesting. Then, to investigate the mechanism they aim to go one step beyond and perform a series of co-IP and IP experiments where they claim that both proteins interact directly. I have several questions on this:

-in Fig 4A the IgG control of the IP is only done with the control cas9 but not with SCARB2 cas9. This control seems missing. The same is for Fig 4B

This is very interesting as it would mean that SCARB2 could be a major regulator of MYC function. Mainly, this finding would mean that a protein and receptor that is supposed to be in the membrane of lysosomes where it regulates lysosomal/endosomal transport binds to a nuclear transcription factor in the nucleus. In that regard the stainings of SCARB2 and MYC in Figure 5 D and F look strange when considering localization: on one hand Myc seems in cytoplasm while on the other hand SCARB2 seems exclusively localized in the nucleus, while one would expect the majority of it in the cytoplasm and co-localizing with lysosomes. Have the authors confirmed the specificity of their antibodies? If correct, the latter would mean that this receptor exits the lysosome to enter in the nucleus and bind MYC, do they see double localization?

Or does it bind to cytoplasmic MYC?

Taking into account the potential impact of that finding, some additional confirmations that the stainings are correct and that the majority of the SCARB2 protein is in the nucleus and not in lysosomes seems needed.

Minor points

1)line 175: remove "obviously"

Reviewer #3:

Remarks to the Author:

In this study, Wang et al., have investigated the functional and clinical significance of SCARB2 in regulation of cancer stemness in hepatocellular carcinoma. First, they have initiated a with a metabolic gene CRISPR/Cas9 knockout screen in HCCM3-derived tumor spheres based on their sizes. Suppression of SCARB2 in HCCM3, HepG2 and primary organoid led to suppression of cancer stemness properties including self-renewal, migration and drug resistance. They further validated their findings in vivo using Scarb2 lineage traced mice with treatment of DEN and showed that Scarb2+ cells are more capable to initiate tumor formation. Furthermore, they demonstrated that Cre-mediated recombination with oncogenic MYC expression in Scarb2+ cells to drive HCC tumor formation. In clinical samples, SCARB2 is significantly correlated with MYC. Mechanistically, SCARB2 was found to bind with MYC, which facilitates MYC acetylation by interfering with HDAC3-mediated MYC deacetylation. As a result, MYC activity was enhanced. To explore the therapeutic potential of targeting SCARB2 protein, they did a screen with FDA-approved drugs via molecular docking approach and found that Polymyxin B binds well with SCARB2 and demonstrated significant tumor suppressive effect in cell-based and PDX models. Thus far, the novel of SCARB2 in regulation of cancer stemness has not been reported before. This study provides a mechanistic insight to target liver CSC vulnerability via targeting SCARB2-mediated metabolic pathways. Please find my comments below for further improvement of this manuscript:

1. SCARB2 was identified via a human CRISPR metabolic gene knockout library in HCC cells HCCM3. I am wondering whether each sgRNA is well presented in each tumor cells. I am doubtful about validity of the data. The authors should show the plot showing the normalized read count of SCARB2 between large size and small size. Also, it will be more appropriate to compare spheres vs differentiated progenies.
2. The use of different HCC cell lines, organoid, clinical samples, mouse samples were not consistent throughout the study. For example, the human tumor organoid apoptosis finding presented in Figure 1f, and organoid was not evaluated for cell growth, migration and self-renewal etc.
3. Did the authors co-stain with other known liver CSC markers?
4. The authors did scRNA-seq on human primary HCC tumorspheres. Did they show that SCARB2-high samples showed enhanced MYC signature genes upon pathway analysis?
5. Did the authors check whether SCARB2 is associated with drug resistance (eg. sorafenib using some publicly available datasets)?
6. For the mechanistic part, how the authors exclude the possibility that SCARB2 regulates cancer stemness via CD36-mediated lipid metabolism?
7. Any potential off-target effects of Polymyxin B?

8. Can Polymyxin B suppress the liver CSC subset? Evaluation of well-known liver CSC markers should be evaluated upon treatment.

Point-by-point response

Reviewer #1 - MYC regulation - (Remarks to the Author):

In this manuscript, Wang et al., report a role for the lysosomal membrane protein SCARB2 in driving cancer stem cells in hepatocellular carcinoma (HCC). They conduct a focused CRISPR knockout screen in human HCC tumorspheres and identify a number of genes required for their maintenance, one of which is SCARB2. They show that SCARB2 is overexpressed in HCC, and that its overexpression correlates with poor patient survival. They also show that Scarb2 deletion suppresses tumor initiation and progression in genetically engineered mouse models of liver cancer. By transcriptional profiling, they show that canonical Myc target gene sets are suppressed in the Scarb2 null setting, and present evidence that loss of Scarb2 is associated with decreased binding of Myc to chromatin, as well as decreased acetylation of Myc on lysine 148 (K148). They further present evidence that SCARB2 promotes acetylation of MYC at K148 by antagonizing the effects of HDAC3. By in silico screening, they identify the FDA-approved drug polymyxin B (PMB; usually used to treat conjunctivitis) as an inhibitor of the SCARB2–MYC interaction, and show that it has anti-tumor activity in vivo in combination with sorafenib.

This is a very interesting and potentially very important study. The identification of SCARB2 as a driver of cancer stem cells in HCC is an important finding, as are the mechanistic connections to MYC, and identification of PMB as an agent with significant anti-tumor activity in vivo. It is easy to see how these findings could lead to a new way to target MYC (a prized but undruggable drug target) in liver cancer. The work itself is quite exhaustive and comprehensive, of generally very high quality, and interpreted fairly. Key conclusions are backed up by multiple layers of evidence, and it is difficult to find fault with most of the approaches or the conclusions drawn from the totality of the work that is presented.

That said, one area that should be stronger, and needs to be strengthened, are the mechanistic connections between SCARB2, HDAC3, and MYC.

1. In Figure 3, the authors show that *Scarb2* deletion suppresses *Myc* target genes in mice. But the accompanying western blot (Fig. S3b) shows that *Myc* levels are lower in these mice. Rather than being that *Scarb2* controls the binding of *Myc* to chromatin, as they later show from cell based assays, this could mean that *in vivo* *Scarb2* controls *Myc* levels. This discrepancy raises the concern that *Scarb2* is acting differently *in vitro* and *in vivo* and as *Myc* levels could simply change as a result of the decreased proliferation of the cancer cells, this raises concern that the mechanistic model built from cell line studies is incorrect.

Re: Thank you for your careful observation. Actually, we assessed the statistical significance of *Myc* protein levels in HCC cells of *Cre^{Alb}Scarb2^{F/F}Myc* mice (n = 3) and *Cre^{Alb}Myc* mice (n = 3) by Student's t-test. There was no statistical difference ($p = 0.3820$) in the MYC protein level between these two groups (Below panel, Revised Supplementary Fig 3b). To further confirm whether *Scarb2* deletion affects MYC level, we detected the MYC protein level of HCC tissue samples from more *Cre^{Alb}Scarb2^{F/F}Myc* mice (n = 4) and *Cre^{Alb}Myc* mice (n = 3). Similar with previous observation, lack of *Scarb2* indeed showed no effects on the MYC expression (Below panel A). Moreover, we performed ChIP-qPCR assay to detect the binding of MYC to several target genes in HCC cells from *Cre^{Alb}Scarb2^{F/F}Myc* mice and *Cre^{Alb}Myc* mice. *Scarb2* deletion *in vivo* decreased MYC binding to chromatin (Below panel, Revised Fig 3p). Therefore, *Scarb2* deletion *in vivo* interfered with MYC binding to target genes, but did not affect the protein level of MYC.

Supplementary Fig 3b

A

(A) Expressions of MYC and SCARB2 were detected by Western blotting in the HCC cells from *Cre^{Alb}Myc* mice (n=3) and *Cre^{Alb}Scarb2^{F/F}Myc* mice (n=4).

2. Figure 3 also shows that SCARB2 deletion decreases the binding of MYC to chromatin, and this then becomes the proposed mechanism of action. They go on to claim that this is due to control of acetylation of MYC at K148 (by interfering with HDAC3 interaction with MYC). But they never show that the K148R mutant of MYC has altered chromatin binding properties (hard to imagine given how far away this residue is from the DNA binding domain of MYC); they do not show that SCARB2 and HDAC3 directly affect MYC binding to chromatin; they do not show that PMB alters MYC binding to chromatin. Indeed, they argue that SCARB2 influences the interaction of MYC with chromatin co-factors—not proteins that impact MYC binding)—creating a confusing hole in their argued mechanism of action. Further, there is no evidence that SCARB2 or HDAC3 interacts directly with MYC, which creates further weaknesses in the model.

Re: Follow your suggestion, we performed ChIP-qPCR assays to detect the effect of MYC K148R mutation, *Scarb2* knockout, HDAC3 inhibition or PMB treatment on the MYC binding to its target genes in revised MS. We found that K148R mutant of MYC

displayed decreased binding to target genes (Below panel, Revised Fig 4h). Muto et al. also reported that acetylation of K148 increased MYC activity without affecting its protein abundance in myeloid malignancies (Muto et al., 2022). However, the mechanistic basis for increased MYC function upon acetylation of K148 remains unknown. We performed the preliminary exploration of the potential mechanism by which K148 acetylation affected MYC binding to chromatin. The DNA binding and transcriptional activity of MYC requires its dimerization with MAX (Lourenco et al., 2021). We used Co-IP assay and Proximity Ligation Assay (PLA) to identify whether *MYC K148R* mutation affected its dimerization with MAX. As indicated in below panel A and B, K148R mutation indeed decreased its interaction with MAX and the formation of MYC/MAX heterodimer. We speculated that K148 acetylation of MYC possibly altered the three-dimensional or spatial structure of MYC protein, making it easier to form the heterodimer of MYC/MAX. This point needs to be confirmed by the co-crystallization of full-length MYC/MAX with E-box complex. We also discussed this part in the revised MS (Page 16, line 388-393).

Fig 4h

(A) K148R mutation of MYC decreased the interaction of MYC and MAX. Cellular extracts were IP with rabbit immunoglobulin G (IgG) and anti-MYC Ab and blotted with anti-MAX Ab. **(B)** Colocalization of MYC and MAX was detected in *MYC^{WT}* and *MYC^{K148R}* HCCLM3 cells by the Duolink PLA assay. Scale bar, 5 μ m. Data are represented as means \pm SEM. Statistical significance was determined by two-tailed Student's t test.

In addition, *SCARB2* knockout or PMB treatment decreased the MYC binding to chromatin, while HDAC3 inhibition showed the opposite effect (Below panels, Revised Fig 3o, Fig 5i and Fig 6j). Similar with K148R mutation, *SCARB2* knockout decreased MYC/MAX interaction and the formation of MYC/MAX heterodimer (Below panels, Revised Fig 3q and 3r), suggesting that *SCARB2* affect MYC binding to chromatin.

Moreover, we used purified system to detect the direct interaction of SCARB2 or HDAC3 with MYC *in vitro*, and found that SCARB2 or HDAC3 could interact directly with MYC (Below panels, Revised Supplementary Fig 4g and 4h).

Fig 3o

Fig 5i

Fig 6j

Fig 3q

Fig 3r

Supplementary Fig 4g

Supplementary Fig 4h

Reviewer #2 - HCC initiation, CSCs (Remarks to the Author):

In this manuscript by Wang et al., the authors aim at identifying drivers of HCC cancer stem cell development and potential therapeutics that eliminate HCC. For that, the authors perform a metabolic CRISPR/Cas9 screen and identify Scarb2 KO reduces HCC development. They suggest that Scarb2 maintains CSC state. To proof that mechanistically, they go one step further and describe that the mechanism relates to SCARB2 binding to MYC which inhibits MYC deacetylation in a HDCA3-dependendent manner and subsequently triggers MYC activation.

The manuscript is a great effort, a tour the force to try to investigate the role of this membrane lysosomal protein in HCC. The authors use all types of in vitro models, from spheroids to tumour organoids and cell lines as well as in vivo mouse models. While there is a lot of work, my main concerns lie on the conclusion that SCARB2 is important for cancer stem cells, which is mainly drawn from the use of cancer cell lines that have been long term expanded in culture., or from a mouse model that deletes the gene in development. The human data is nice but correlative. In addition, the interaction with MYC seems interesting but poses questions. My main comments are below.

Major comments

1. SCARB2 as a critical gene for maintaining the stemness of HCC cells

the main conclusion that SCARB2 plays a role in HCC CSC viability comes from the screen in cell lines. While this reviewer agrees that this protein plays a role in these cell lines, not proven that this is through an effect on CSC. There are some conclusions here that do not seem to match between the cell lines and the primary tissue, or tumor organoids used in figure 1J. In fact cancer cell lines are homogenous and, contrary to organoids, would not contain CSC and differentiated cells. This conclusion could only be obtained if the authors can identify CSC in their cultures and then compare them to non-CSC and observe that SCARB2 plays different roles in the two populations. Since SCARB2 mutant organoids are nicely viable (according to the live-death staining) and only when treated with Sorafenib they die, that would rather suggest the opposite, that SCARB2 KO affects the non-proliferative non-CSC pool in organoids. One potential experiment to address that would be that the authors stain the WT and mutant

untreated organoids for CSC markers and observe a reduction of these. This could also be done by FACS for more quantitative analysis. In addition proof that it is the CSC pool in the cell lines the one affected would be necessary to sustain that claim.

Re: Thank you very much for your constructive suggestion. As for your first concern whether CSCs exist in the HCC cell lines, we used CD133 and CD13 as liver CSCs (LCSCs) surface markers to examine the proportion of CD13+CD133+ in HCC cells by flow cytometry. Similar with previous study (Shi et al., 2022), the proportion of CD13+CD133+ is 1.19% in HCCLM3 cells, 1.01% in HepG2, and 5.09% in primary HCC cells (Below Panel, Supplementary Fig 1h), suggesting that there exists CSCs in our culture system. We then sorted CD13+CD133+ and CD13-CD133- subpopulations from these HCC cells. SCARB2 was identified more strongly expressed in CSCs (CD13+CD133+) than that in non-CSCs (CD13-CD133-, Below Panel, Revised Fig 1h). When SCARB2 was knocked down in liver CSCs, the expression of stem markers (*CD24* and *EpCAM*) and stem transcription factors (*Nanog*, *SOX2*, and *OCT4*) were reduced (Below Panel, Revised Supplementary Fig 1i). *SCARB2* deficiency showed strong inhibition of proliferation in CSCs, but had slight effect on non-CSCs (Below Panel, Revised Fig 1i). Similar with previous observation, *SCARB2* knockout in liver CSCs suppressed their capacity of sphere formation (Below Panel, Revised Fig 1j). These data suggest that *SCARB2* plays different roles in the liver CSCs and non-CSCs.

Supplementary Fig 1h

Fig 1h

Supplementary Fig 1i

Fig 1i

Fig 1j

Following your suggestion, we examined the expression of CSC markers in tumor organoids with or without *SCARB2* knockout by FACS. The proportion of CD24, EpCAM, CD13, or CD133 positive cells was decreased in tumor organoids with *SCARB2* knockout (Below Panel, Revised Fig 1o). In addition, *SCARB2* knockout in HCCLM3 cells decreased the proportion of CD24, EpCAM, CD13, or CD133 positive cells (Below Panel, Revised Fig 1g). These data indicated that *SCARB2* acted as a critical gene for maintaining the stemness of HCC cells.

Fig 1o

Fig 1g

Similarly, the mouse data (depletion of *SCARB2* in *AlbCre* mice) indicates that there is a reduction in tumour size in these mutants. However, the conclusion that this is driven by effect specific on the CSC pool is not proven. Mainly *AlbCre* deletes in the embryo, at then hepatoblast stage, and that is not a "cancer stem cell". In addition,

the model would be artificial, as the deletion would occur in development and not in adulthood, as it is supposed to be for a somatic disease such as cancer. Also, note that the authors insist on mentioning "Suppress, tumour" e.g. in. "Scarb2 deletion suppresses the initiation and progression of HCC." However, the correct wording is reduce tumour growth and tumour metastasis, since the mutant cells and mice develop tumours but these are smaller, or when mutant cells are injected these still generate metastasis, although less (as seen in figure 2D-K), the conclusion that "suppress is. This should be amended through the text.

Mainly there is a correlation between loss of this gene and overall decrease in tumour growth but there is no proof that this is linked to CSC reduction.

Re: We agree that loss of Scarb2 in AlbCre mice mainly decreased tumour growth in the original MS. In the revised MS, we detected the proportion of EpCAM, CD133 or CD24 positive cells in liver cancer tissues from $Cre^{Alb}Myc$ mice and $Cre^{Alb}Scarb2^{F/F}Myc$ mice. The proportion of EpCAM, CD133 or CD24 positive cells significantly decreased in $Cre^{Alb}Scarb2^{F/F}Myc$ mice than that in $Cre^{Alb}Myc$ mice, demonstrating that knockout of Scarb2 had effect on the CSC pool (Below Panel, Revised Fig 2g). In addition, we used extreme limiting dilution assay to assess the tumor repopulation ability of tumor cells from $Cre^{Alb}Myc$, $Cre^{Alb}Scarb2^{F/+}Myc$, and $Cre^{Alb}Scarb2^{F/F}Myc$ mice. As indicated in Revised Fig 2f, Scarb2 deletion reduced the frequency of tumors-initiating cells. Altogether, these data indicated that Scarb2 had effect on the HCC CSC pool.

Fig 2g

Fig 2f

Number of tumour cells	$Cre^{Alb}Myc$	$Cre^{Alb}Scarb2^{F/+}Myc$	$Cre^{Alb}Scarb2^{F/F}Myc$
100,000	10/10	8/10	4/10
5,0000	9/10	6/10	2/10
25,000	9/10	3/10	1/10
12,500	6/10	3/10	1/10
6,250	5/10	1/10	0/10
TIC frequency (95% interval)	1/13506 (1/21063-1/8660)	1/56715 (1/89509-1/35936)	1/206291 (1/413668-1/102875)
P value	$p < 0.0001$		

In Figure 1, SCARB2 was identified more strongly expressed in CSCs, and SCARB2 knockout inhibited the proliferation and capacity of sphere formation of CSCs. We further investigated the role of SCARB2 in liver CSCs *in vivo*. Sphere formation is used to enrich CSCs from HCCLM3 cells (Ma et al., 2019), and these tumor spheroids were

infected with $CTRL^{Cas9}$ and $SCARB2^{Cas9}$ virus particles and subcutaneously inoculated into BALB/c nude mice. $SCARB2$ knockout resulted in significantly decreased tumor growth and reduced tumor sizes and tumor weights (Below Panel, Revised Fig 2o-2q and Supplementary Fig 2j). At the endpoint of inoculation days, the tumors generated from $SCARB2$ knockout spheroids showed the decreased proportions of CD24, EpCAM, CD13, or CD133 positive cells (Below Panel, Revised Fig 2r).

Fig 2o

Fig 2p

Fig 2q

Supplementary Fig 2j

Fig 2r

In addition, we have corrected the description with “*Scarb2* deletion reduced tumour growth and tumour metastasis” throughout the revised manuscript.

2. Many figures suffer from lack of information on number of biological and technical replicates used. Mainly, except for the number of human samples in Fig 1 and mouse experiments this is missing in all in vitro experiments.

In addition, the statistical analysis is generalized on a methods section in the manuscript. Unfortunately, this precludes the evaluation whether the statistical tests are correct. Mainly, the analyses of all figures where there are multiple biological replicates with multiple technical replicates should follow a t-test of means if data is normally distributed, which has not been addressed here. I suggest authors to follow the directions published by Lord et al., J. Cell Biol. 2020 Vol. 219 No. 6 or Pollard et

al., *Molecular Biology of the cell* Volume 30 June 1, 2019 among others.

Re: Following your suggestion, we supplemented the information on number of biological, technical replicates and statistical tests in the figure legend section of revised MS following the directions published by Lord et al (Lord et al., 2020).

3. Myc and SCRAB2 interaction

The authors aim at investigating the mechanism by which myc tumours are reduced in the absence of SCRAB2. They observe a clear correlation between the mutant cells and an increase in MYC targets. This is interesting. Then, to investigate the mechanism they aim to go one step beyond and perform a series of co-IP and IP experiments where they claim that both proteins interact directly. I have several questions on this:

-in Fig 4A the IgG control of the IP is only done with the control cas9 but not with SCRAB2 cas9. This control seems missing. The same is for Fig 4B

Re: Thank you for your careful observation and suggestion. We re-performed the co-IP experiments of Fig 4a and Fig 4b and the data was shown in the below panel and revised Fig. 4a and 4b.

Fig 4a

Fig 4b

This is very interesting as it would mean that SCRAB2 could be a major regulator of MYC function. Mainly, this finding would mean that a protein and receptor that is supposed to be in the membrane of lysosomes where it regulates lysosomal/endosomal transport binds to a nuclear transcription factor in the nucleus. In that regard the stainings of SCRAB2 and MYC in Figure 5 D and F look strange when considering localization: on one hand Myc seems in cytoplasm while on the other hand SCRAB2 seems exclusively localized in the nucleus, while one would expect the

majority of it in the cytoplasm and co-localizing with lysosomes. Have the authors confirmed the specificity of their antibodies? If correct, the later would mean that this receptor exits the lysosome to enter in the nucleus and bind MYC, do they see double localization?

Or does it bind to cytoplasmic MYC?

Taking into account the potential impact of that finding, some additional confirmations that the stainings are correct and that the majority of the SCARB2 protein is in the nucleus and not in lysosomes seems needed.

Re: Following your suggestion, we re-confirmed the specificity of SCARB2 antibody (Abcam, ab176317) and MYC antibody (R&D, MAB3696) in HepG2 cells with or without SCARB2 or MYC depletion. As shown in the below panel A, when SCARB2 or MYC was depleted in HepG2 cells, the staining of SCARB2 or MYC was lack, suggesting the specificity of the anti-SCARB2 and MYC antibodies. In addition, we confirmed that MYC and SCARB2 both localized in the cytoplasm and nucleus of HepG2 cells (Below Panel, Revised Fig 5d). In addition, we co-stained SCARB2 and marker of lysosomes LAMP2 in HCCLM3 cells, and found that SCARB2 also localized in nucleus except for co-localization with LAMP2 in the cytoplasm (Below Panel B).

Legend: (A) Cellular localization of MYC or SCARB2 in HepG2 cells with or without SCARB2 depletion or MYC depletion was detected with immunostaining. Scale bar, 5 μm. (B) Colocalization of SCARB2 and LAMP2 in HCCLM3 cells was detected with immunostaining. Scale bar, 5 μm.

Minor points

1) line 175: remove “obviously”

Re: We have already corrected it in the revised MS (Page 10, Line 222).

Reviewer #3 - HCC CSCs, metastasis, resistance (Remarks to the Author):

In this study, Wang et al., have investigated the functional and clinical significance of SCARB2 in regulation of cancer stemness in hepatocellular carcinoma. First, they have initiated a with a metabolic gene CRISPR/Cas9 knockout screen in HCCM3-derived tumor spheres based on their sizes. Suppression of SCARB2 in HCCM3, HepG2 and primary organoid led to suppression of cancer stemness properties including self-renewal, migration and drug resistance. They further validated their findings in vivo using Scarb2 lineage traced mice with treatment of DEN and showed that Scarb2+ cells are more capable to initiate tumor formation. Furthermore, they demonstrated that Cre-mediated recombination with oncogenic MYC expression in Scarb2+ cells to drive HCC tumor formation. In clinical samples, SCARB2 is significantly correlated with MYC. Mechanistically, SCARB2 was found to bind with MYC, which facilitates MYC acetylation by interfering with HDAC3-mediated MYC deacetylation. As a result, MYC activity was enhanced. To explore the therapeutic potential of targeting SCARB2 protein, they did a screen with FDA-approved drugs via molecular docking approach and found that Polymyxin B binds well with SCARB2 and demonstrated significant tumor suppressive effect in cell-based and PDX models. Thus far, the novel of SCARB2 in regulation of cancer stemness has not been reported before. This study provides a mechanistic insight to target liver CSC vulnerability via targeting SCARB2-mediated metabolic pathways. Please find my comments before for further improvement of this manuscript:

1. SCARB2 was identified via a human CRISPR metabolic gene knockout library in HCC cells HCCLM3. I am wondering whether each sgRNA is well presented in each tumor cells. I am doubtful about validity of the data. The authors should show the plot showing the normalized read count of SCARB2 between large size and small size. Also, it will be more appropriate to compare spheres vs differentiated progenies.

Re: We analyzed the normalized read counts of all sgRNAs in HCCLM3 cells infected with CRISPR/Cas9 metabolic gene knockout library prior to sphere-forming culture. Our analysis revealed that around 98% of all sgRNAs retained in the HCCLM3 mutant pool (below panel, Revised Fig 1b). However, some HCCLM3 mutant cells were unable to form tumor spheres during the sphere-forming culture process. Additionally,

tumor spheres ranging in size from 40 μm to 70 μm were excluded from our analysis. As a result, the genes enriched in the remaining large and small tumor spheres did not cover full representation of the pooled CRISPR/Cas9 metabolic gene library. Following your suggestion, we supplemented the normalized read counts of all *SCARB2* sgRNAs in Revised Fig 1f (below panel), which demonstrated a significant increase in *SCARB2* targeting sgRNAs in small tumor spheres.

We agree with this reviewer that it is appropriate to identify genes specially playing critical role on CSCs but not non-CSCs by comparing the differential gRNA enrichment in spheres vs differentiated progenies. At the beginning of this study, we focused on the enriched sgRNAs in smaller spheres and further validated each of the top 10 genes enriched in small tumorspheres. Most screened sgRNAs that targeted these 10 genes decreased the tumorsphere formation capacity of HCC cells, suggesting this method is suitable for screening target gene supporting tumorspheres formation capacity. As reviewer 2 proposed, we also compared the role of *SCARB2* in CSC and non-CSCs. Lack of *SCARB2* knockout showed strong inhibition of proliferation in CSCs, but had slight effect on non-CSCs, suggesting *SCARB2* plays important role in CSCs proliferation but not in non-CSCs (Revised Fig 1i).

2. The use of different HCC cell lines, organoid, clinical samples, mouse samples were not consistent throughout the study. For example, the human tumor organoid apoptosis finding presented in Figure 1f, and organoid was not evaluated for cell growth, migration and self-renewal etc.

Re: Following your suggestion, we performed CCK-8 and invasion assays on human tumor organoids. SCARB2 depletion decreased the growth and invasion of tumor organoids (Below panel, Revised Fig 1m and 1n).

Fig 1m

Fig 1n

3. Did the authors co-stain with other known liver CSC markers?

Re: Following your suggestion, we co-stained SCARB2 with other known liver CSC markers in HCCLM3 cells and primary HCC cells. The SCARB2 positive liver cancer cells co-expressed liver CSC markers including CD13, CD133, CD24 or EpCAM. These data were shown below and indicated in revised Supplementary Fig1g.

Supplementary Fig 1g

4. The authors did scRNA-seq on human primary HCC tumorspheres. Did they show that SCARB2-high samples showed enhanced MYC signature genes upon pathway analysis?

Re: Following your suggestion, we analyzed the enrichment of MYC signature genes in SCARB2-negative and SCARB2-positive cells by gene set enrichment analysis (GSEA). The SCARB2 positive cells exhibited the enrichment of MYC target genes. These data were shown below and indicated in revised Supplementary Fig 3c.

Supplementary Fig 3c

5. Did the authors check whether SCARB2 is associated with drug resistance (eg. sorafenib) using some publicly available datasets?

Re: Following your suggestion, we analyzed the levels of SCARB2 in parental cells and sorafenib-resistant HCC cells in GEO dataset (GSE121153). SCARB2 expression was upregulated in sorafenib-resistant HCC cells (below panel, Revised Supplementary Fig 1m), suggesting SCARB2 may be associated with sorafenib resistance in HCC.

Supplementary Fig 1m

6. For the mechanistic part, how the authors exclude the possibility that SCARB2 regulates cancer stemness via CD36-mediated lipid metabolism?

Re: Actually, we had detected the expression of CD36 in SCARB2 knockout HCC cells, and found that SCARB2 deletion had no effect on the CD36 expression (below panel A). However, SCARB2 knockout decreased the lipid accumulation in HCC cells (below panel B), which may be associated with the role of SCARB2 in cholesterol transport (Heybrock et al., 2019). It needs substantial work to confirm whether and how SCARB2-mediated lipid metabolism regulates cancer stemness. In this study, we focused on exploring the effect of SCARB2 on cancer stemness by regulating MYC activity. Therefore, these data are shown below for your reviewing and will not display in the revised MS.

(A) The proportion of CD36 positive cells in CTRL^{Cas9} or SCARB2^{Cas9} HCCLM3 cells was analyzed by Flow cytometry. (B) Lipid droplets were stained with Bodipy^{493/503} (Invitrogen). Cell suspensions from CTRL^{Cas9} or SCARB2^{Cas9} HCCLM3 cells were incubated with 0.2 µg/mL Bodipy^{493/503} solution in the dark for 30min at 37°C. Then the stained cells were analyzed by Flow cytometry and the MFI (Mean Fluorescence Intensity) of BODIPY^{495/503} probe was calculated by FlowJo 10.8.1 software.

7. Any potential off-target effects of Polymyxin B?

Re: We performed CCK-8 and colony formation assays to detect the effect of Polymyxin B on MYC- or SCARB2- knockdown HCC cells in the original MS (Below panel, Fig 6p and 6q). MYC or SCARB2 depletion diminished the anti-proliferation and anti-stemness ability of PMB treatment, suggesting that Polymyxin B has little off-target effects in HCC cells *in vitro*. However, it is hard to say there is not any off-target effects of Polymyxin B *in vivo*. The systemic treatment of Polymyxin B killed bacteria (Alipour et al., 2008), induced necrosis of macrophages (Kagi et al., 2022), and decreased the

regulatory T cells (Treg) population (Cappelli et al., 2012), which may contribute to the anti-tumor activities of Polymyxin B *in vivo*. We have discussed this point in the revised MS (Page 17, line 407-412).

Fig 6p

Fig 6q

8. Can Polymyxin B suppress the liver CSC subset? Evaluation of well-known liver CSC markers should be evaluated upon treatment.

Re: We examined the proportion of CD133, CD13, EpCAM, and CD24 positive cells in human primary HCC cells after treatment with PMB. PMB significantly decreased these liver CSC markers of HCC cells, indicating that PMB could suppress the liver CSC subsets (Below panel, Supplementary Fig 5e).

Supplementary Fig 5e

References

Alipour, M., Halwani, M., Omri, A., and Suntres, Z.E. (2008). Antimicrobial effectiveness of liposomal polymyxin B against resistant Gram-negative bacterial strains. *Int J Pharm* 355, 293-298.

Cappelli, C., Lopez, X., Labra, Y., Montoya, M., Fernandez, R., Imarai, M., Rojas, J.L., Miranda, D., Escobar, A., and Acuna-Castillo, C. (2012). Polymyxin B increases the depletion of T regulatory cell induced by purinergic agonist. *Immunobiology* 217, 307-315.

Heybrock, S., Kanerva, K., Meng, Y., Ing, C., Liang, A., Xiong, Z.J., Weng, X., Ah Kim, Y., Collins, R., Trimble, W., *et al.* (2019). Lysosomal integral membrane protein-2 (LIMP-2/SCARB2) is involved in lysosomal cholesterol export. *Nat Commun* 10, 3521.

Kagi, T., Naganuma, R., Inoue, A., Noguchi, T., Hamano, S., Sekiguchi, Y., Hwang, G.W., Hirata, Y., and Matsuzawa, A. (2022). The polypeptide antibiotic polymyxin B acts as a pro-inflammatory irritant by preferentially targeting macrophages. *J Antibiot (Tokyo)* 75, 29-39.

Lord, S.J., Velle, K.B., Mullins, R.D., and Fritz-Laylin, L.K. (2020). SuperPlots: Communicating reproducibility and variability in cell biology. *J Cell Biol* 219.

Lourenco, C., Resetca, D., Redel, C., Lin, P., MacDonald, A.S., Ciaccio, R., Kenney, T.M.G., Wei, Y., Andrews, D.W., Sunnerhagen, M., *et al.* (2021). MYC protein interactors in gene transcription and cancer. *Nat Rev Cancer* 21, 579-591.

Ma, X.L., Sun, Y.F., Wang, B.L., Shen, M.N., Zhou, Y., Chen, J.W., Hu, B., Gong, Z.J., Zhang, X., Cao, Y., *et al.* (2019). Sphere-forming culture enriches liver cancer stem cells and reveals Stearoyl-CoA desaturase 1 as a potential therapeutic target. *BMC Cancer* 19, 760.

Muto, T., Guillamot, M., Yeung, J., Fang, J., Bennett, J., Nadorp, B., Lasry, A., Redondo, L.Z., Choi, K., Gong, Y., *et al.* (2022). TRAF6 functions as a tumor suppressor in myeloid malignancies by directly targeting MYC oncogenic activity. *Cell Stem Cell* 29, 298-314 e299.

Shi, J., Guo, C., Li, Y., and Ma, J. (2022). The long noncoding RNA TINCR promotes self-renewal of human liver cancer stem cells through autophagy activation. *Cell Death Dis* 13, 961.

Reviewers' Comments:

Reviewer #1:

Remarks to the Author:

In this revised manuscript, Wang et al., have responded in a substantive way to all of the original criticisms raised by this reviewer in the primary round of critiques. I am particularly pleased to see that the mechanistic connections between MYC–HDAC3–SCARB2 have been fleshed out, which has gone a long way to increase the impact of the study. I am convinced the authors have separated the effects of SCARB2 disruption on DNA binding by MYC from any effects on MYC protein levels, which was a major concern. I also conclude that the inclusion of the new data on the K148R mutant of MYC is particularly important, and the argument that HDAC3 and SCARB2 descend on this residue of MYC to control DNA binding is now quite compelling. I still find it odd that a mutation in the MYC transactivation domain would effect interaction with MAX, but the data are convincing, which in the end just acts to increase the interest in the story. Overall, this is an important and timely contribution.

Reviewer #2:

Remarks to the Author:

The manuscript is much improved and the authors have addressed well some of my concerns. It is clear that Scarb2 depletion induces tumour growth and metastasis and promotes tumour initiation in the models tested. It is also clearly demonstrated that the effect is through its interaction with MYC. However, my concern that the observed effect is due to an impact on Cancer Stem Cells remains. Cancer stem cells is a concept whereby one cell behaves as a stem cell (either from healthy or tumour, in the case of CSC, from tumour) and gives rise to daughter cells that have less "stemness" and instead acquire a differentiation state. As explained on the previous version, cancer cell lines grown in 2D or 3D cannot exhibit this stemness/differentiation states as are mostly homogenous and clonal. Patient derived organoids would be helpful, and the authors have made the effort on incorporating these into their "cancer stem cell" analysis, which is loadable and conceptually better. In this revised version the authors use CD133/CD24/Cd13/EpCAM as Cancer stem cell markers of their population. In the liver, these markers are used as markers for ductal cells in healthy liver, and also observed expressed in tumours, however, that these are marking a cancer stem cell population in the cell lines and other models used by these authors is not proven at all in this manuscript. The authors mention that the mutant Scarb2 livers present less of these cells, however, because they deplete using the AlbCre driver that is active during the embryonic pool, at the hepatoblast stage, it is not possible to detangle if the reduction of EpCAM+CD133+/CD24 + cells is due to a reduction in the number of healthy cells that eventually give rise to tumours or in the number of tumour cells that have more stemness potential, as the authors claim. For that claim to be sustained the authors should first, proof that the EpCAM+CD133+ is a cancer stem cell population when compared to EpCAM-CD133- by performing clonogenicity and dilution transplantation assay, and second, then on the pure CSC population then deplete Scarb2 observe reduction of tumorigenesis in these.

I don't expect the authors to do these experiments. They should tone down the claim about cancer stem cells, and maybe restrict it to what the data shows by saying something like "Scarb2 promotes tumour growth or tumour initiating state"

On another note, the authors now present in Fig 2f a dilution assay and xenotransplantation on the mutant cells (not specifically CSC mutants) and however the population still has a 40% (4/10) tumour initiation efficiency. How to explain that if these are Cancer stem cells ?

Reviewer #3:

Remarks to the Author:

The authors have addressed my comments satisfactorily. I have no further comment to the manuscript in this revised version.

Reviewer #1 (Remarks to the Author):

In this revised manuscript, Wang et al., have responded in a substantive way to all of the original criticisms raised by this reviewer in the primary round of critiques. I am particularly pleased to see that the mechanistic connections between MYC–HDAC3–SCARB2 have been fleshed out, which has gone a long way to increase the impact of the study. I am convinced the authors have separated the effects of SCARB2 disruption on DNA binding by MYC from any effects on MYC protein levels, which was a major concern. I also conclude that the inclusion of the new data on the K148R mutant of MYC is particularly important, and the argument that HDAC3 and SCARB2 descend on this residue of MYC to control DNA binding is now quite compelling. I still find it odd that a mutation in the MYC transactivation domain would effect interaction with MAX, but the data are convincing, which in the end just acts to increase the interest in the story. Overall, this is an important and timely contribution.

Re: Thank you very much for your positive and detailed comments. We really appreciate your professional suggestions that has enabled us to make further improvements to our manuscript.

Reviewer #2 (Remarks to the Author):

The manuscript is much improved and the authors have addressed well some of my concerns. IT is clear that Scarb2 depletion induces tumour growth and metastasis and promotes tumour initiation in the models tested. It is also clearly demonstrated that the effect is through its interaction with MYC. However, my concern that the observed effect is due to an impact on Cancer Stem Cells remains. Cancer stem cells is a concept whereby one cell behaves as a stem cell (either from healthy or tumour, in the case of CSC, from tumour) and gives rise to daughter cells that have less “stemness” and instead acquire a differentiation state. As explained on the previous version, cancer cell lines grown in 2D or 3D cannot exhibit this stemness/differentiation states as are mostly homogenous and clonal. Patient derived organoids would be helpful, and the authors have made the effort on incorporating these into their “cancer stem cell” analysis, which is loadable and conceptually better. In this revised version the authors use CD133/CD24/Cd13/EpCAM as Cancer stem cell markers of their population. In the liver, these markers are used as markers for ductal cells in healthy liver, and also observed expressed in tumours, however, that these are marking a cancer stem cell population in the cell lines and other models used by these authors is not proven at all in this manuscript. The authors mention that the mutant Scarb2 livers present less of these cells, however, because they deplete using the AlbCre driver that is active during the embryonic pool, at the hepatoblast stage, it is not possible to detangle if the reduction of EpCAM+CD133+/CD24 + cells is due to a reduction in the number of healthy cells that eventually give rise to tumours or in the number of tumour cells that have more stemness potential, as the authors claim. For that claim to be sustained the authors should first, proof that the EpCAM+CD133+ is a cancer stem cell population when compared to EpCAM-CD133- by performing clonogenicity and dilution transplantation assay, and second, then on the pure CSC population then deplete Scarb2 observe reduction of tumorigenesis in these. I don't expect the authors to do these experiments. They should tone down the claim about cancer stem cells, and maybe restrict it to what the data shows by saying something like “Scarb2 promotes tumour growth or tumour initiating state”

On another note, the authors now present in Fig 2f a dilution assay and

xenotransplantation on the mutant cells (not specifically CSC mutants) and however the population still has a 40% (4/10) tumour initiation efficiency. How to explain that if these are Cancer stem cells?

Re: Thank you very much for your positive comments and professional suggestion. Follow your suggestion, we have toned down the claim about cancer stem cells and described the results like “*Scarb2 deletion reduced tumour growth, tumour initiating state, or cancer stem cell-like properties*” throughout the paper, including Title, Abstract, Introduction, Results, and Discussion in the revised MS.

In Fig 2f, HCC cells isolated from *Cre^{Alb}Myc* mice or *Cre^{Alb}Scarb2^{F/F}Myc* mice but not pure CSC population were performed in the *in vivo* limiting dilution assays (LDA). Thus, we changed the description of Fig 2f in the result section in the revised MS as follows: “*Scarb2* knockout reduced the tumour initiation efficiency of HCC cells”.

Reviewer #3 (Remarks to the Author):

The authors have addressed my comments satisfactorily. I have no further comment to the manuscript in this revised version.

Re: Thank you for your positive comments. We really appreciate your professional suggestions that has enabled us to make further improvements to our manuscript.